# CORE: Benchmarking LLMs' Code Reasoning Capabilities through Static Analysis Tasks

**Danning Xie**[*]
Department of Computer Science
Purdue University
xie342@purdue.edu

**Mingwei Zheng**[*]
Department of Computer Science
Purdue University
zheng618@purdue.edu

**Xuwei Liu**
Department of Computer Science
Purdue University
liu2598@purdue.edu

**Jiannan Wang**
Department of Computer Science
Purdue University
wang4524@purdue.edu

**Chengpeng Wang**
Department of Computer Science
Purdue University
wang6590@purdue.edu

**Lin Tan** [†]
Department of Computer Science
Purdue University
lintan@purdue.eduu

**Xiangyu Zhang**
Department of Computer Science
Purdue University
xyzhang@cs.purdue.edu

## Abstract

Large language models (LLMs) have been widely adopted across diverse domains of software engineering, such as code generation, program repair, and vulnerability detection. These applications require understanding beyond surface-level code patterns: value propagation, control flow, and interdependence between program elements. However, existing benchmarks primarily evaluate end-to-end outcomes, such as whether code is correctly repaired or generated, leaving the models' ability for program semantic reasoning underexplored. This work presents CORE, a high-quality, human-verified benchmark designed to evaluate LLMs on fundamental static analysis tasks. CORE includes 12,553 task instances spanning data dependency, control dependency, and information flow across programs written in C/C++, Java, and Python. To ensure semantic diversity and reasoning complexity, we propose a semantics-aware diverse sampling strategy that selects targets and task instances based on structural coverage and dependency depth. We evaluate 10 mainstream LLMs and show that, while they perform well at identifying dependencies, models still struggle with tasks that require deeper semantic understanding and multi-step reasoning. We further conduct qualitative analyses to uncover key challenges, such as complex control structures and backward dependency patterns, offering insights into improving LLMs' code reasoning capabilities.

---

[*]The first two authors contributed equally.
[†]Corresponding author.

39th Conference on Neural Information Processing Systems (NeurIPS 2025) Track on Datasets and Benchmarks.

# 1  Introduction

Large Language Models (LLMs) have shown remarkable capabilities across a wide range of domains [1, 2, 3, 4, 5, 6, 7]. In the area of software engineering, they are increasingly used in tasks such as program synthesis [8, 9], program repair [10, 11], and test generation [12, 13]. LLMs are usually prompted with high-level objectives [10, 11], such as identifying a buggy line, generating test cases to reach a target condition, or producing patches to fix faulty behavior.

***Our Motivation of Benchmarking.*** Success in such tasks requires more than surface-level pattern recognition. Specifically, it requires a deep understanding of the program semantics: *how values propagate through statements, how control structures govern program execution, and how different parts of the program influence one another*. For instance, in the fuzzing scenario shown in Figure 1b, to trigger a vulnerability at line 12, the input must satisfy a nested series of conditions at lines 2, 4, 7, 9, and 10. Failing any of these checks leads to early termination or divergence from the path. This kind of control-dependent reasoning is essential for success in these tasks.

At the same time, researchers are beginning to use LLMs as static analyzers, applying them to downstream tasks such as vulnerability detection [14], automatic debugging [15], and program repair [16]. These approaches, no matter whether implicitly relying on or explicitly prompting LLMs for code reasoning, assume that the model has mastered core code semantic analysis skills, which can support downstream clients with exceptional performance empirically. However, their underlying ability to reason about program semantics remains under-evaluated despite promising end-task results. Therefore, we urgently require an effective benchmarking solution to directly assess whether LLMs possess the deep semantic understanding and reasoning capabilities necessary to support complex software engineering tasks.

***Limitations of Existing Benchmarks.*** Existing benchmarks mainly target the evaluation of LLMs upon code-centric tasks in an end-to-end fashion [17, 18, 19, 20, 21], without fundamentally assessing the model's reasoning ability over program semantics. For example, program repair benchmarks such as SWE-Bench [22] and SWE-Lancer [23] provide buggy-fixed code pairs to evaluate patch generation correctness, yet lack fine-grained ground-truth explanations of why each bug occurs. Other efforts that target fundamental program properties rather than downstream applications, like dynamic trace prediction [24, 25, 26], focus exclusively on observed runtime behaviors under specific inputs, leaving unexercised branches and other possible inputs overlooked. Consequently, they fail to provide adequate evaluation of whether LLMs possess the semantic reasoning required for advanced coding tasks, which highlights the need for a benchmark that directly evaluates core program analysis skills.

***Our Benchmark.*** To fill this gap, we introduce CORE, **a high-quality, multi-lingual benchmark** for evaluating LLMs' Code Reasoning capabilities with **fundamental static analysis tasks**. CORE includes human-verified task instances covering *data dependency*, *control dependency*, and *information flow* across C/C++, Java, and Python. The benchmark consists of 12,553 task instances drawn from 180 programs, selected through *semantics-aware diverse sampling* to ensure both **task diversity** and **non-trivial reasoning complexity**. Due to the inability of existing program analysis techniques to automatically and accurately extract comprehensive semantic properties from multi-languages, we adopt a semi-automated annotation pipeline supplemented with substantial human effort. This ensures the soundness and completeness of the annotations, resulting in a rigorously curated, high-quality dataset.

***Empirical Results and Findings.*** Our extensive evaluation of 10 state-of-the-art LLMs, including 6 top reasoning models, along with qualitative analyses, reveals the key following:

- Reasoning models consistently outperform non-reasoning models across most tasks, with a performance margin of 5.2–31.5%; Gemini 2.5 Pro achieves the best overall results.
- Most of the models perform well on simpler tasks of identifying a dependency between two program elements, achieving F1 scores ranging from 68.9% to 92.56%.
- Models struggle with tasks requiring deeper semantic understanding and multi-step reasoning, such as trace generation and source enumeration, with scores falling below 50%.
- Performance drops significantly in the presence of complex control structures, longer function bodies, and backward or non-sequential dependency patterns, with a performance gap up to 49.5%.
- When prompted to provide both classification and a trace, models are more likely to predict the existence of a dependency, indicating increased false positives under greater task complexity.

```
1  // untrusted input          1  int type, len, i = 0;        1  if (codec != null)
2  String header =             2  while (i < n) {              2    in = new LineReader
     getHeader("Content-Type"); 3    type = a[i];                    (codec.InputStream(file), job);
3  // tainted expression       4    if (type == 0) {          4  else {
4  String expr = "%{"+header+"}"; 5      i++; continue;          5    if (start != 0) {
5  // dangerous sink           6    }                         6      skipFirstLine = true;
6  evaluateOGNL(expr);         7    if (i + 1 >= n) return;   7      --start;
                               8    len = a[i + 1];           8      f.seek(start);
                               9    if (i + len > n) return;  9    }
                               10   if (type == 1)            10   in = new LineReader(f, job);
                               11     // vulnerable sink      11 }
                               12     memcpy(out, a + i + 2, len); 12 if (skipFirstLine)
                               13   i += len + 2;             13   start += in.read();  // NPE
                               14 }
       (a) Taint Analysis            (b) Fuzzing                    (c) Fault Localization
```

Figure 1: Real-world motivating examples for data dependency, control dependency, and information flow in security and software engineering applications.

*Availability.* We release the dataset and code at `https://corebench.github.io/` with Apache-2.0 license.

## 2  Background and Motivation

Understanding program semantics requires reasoning beyond surface-level syntax. In traditional program analysis, *dependencies* are central to modeling how specific program values flow between different program lines, facilitating a wide range of applications, including bug detection[27, 28, 29, 30], program optimization [31, 32, 33, 34], and testing[35, 36, 37, 38, 39, 40]. In this work, we evaluate LLMs on three core reasoning tasks: **data dependency**, **control dependency**, and **information flow**. The first two represent fundamental program dependencies, while the third captures higher-level semantic reasoning about how program values propagate through execution. Evaluating LLMs on them provides insight into their ability to capture deep semantic structure in code.

Before introducing the definitions of the three dependencies and their applications, we define several basic units used throughout our benchmark.

**Variables.** A variable is denoted as $name_{line}$, where name is the identifier and line indicates the line number where the value of the variable is assigned or updated. For example, in the statement x = y + 1 on line 3, we refer to variable x as $x_3$, since it is assigned a new value.

**Line Numbers.** We use $\ell_i$ to refer to the $i$-th line of code. This notation is used when describing relationships between statements based on their locations, particularly in control dependency.

**Variable Relationships.** We use $\rightarrow$ to denote a direct relationship between two variables or statements, including data dependencies ($\rightarrow_D$), control dependencies ($\rightarrow_C$), and information flows ($\rightarrow_I$). A data dependency trace is written as a sequence such as $x_1 \rightarrow_D y_3 \rightarrow_D z_5$, where each arrow indicates that the target variable is directly data-dependent on the source. We use $\rightsquigarrow$ to denote a transitive relationship, indicating that one variable is indirectly influenced by another through one or more intermediate steps. For example, the above trace implies $x_1 \rightsquigarrow_D z_5$.

### 2.1  Data Dependency

A *data dependency* occurs when the value of one variable depends on the value of another, typically arising when variables are assigned and then subsequently used [41].

```
1  x = 5
2  y = x + 2
3  z = y + 3  # z depends on y, and transitively on x
```

Here, $y_2$ is directly data dependent on $x_1$, and $z_3$ is directly data dependent on $y_2$. This forms a transitive data dependency trace: $x_1 \rightarrow_D y_2 \rightarrow_D z_3$. Therefore, $z_3$ has an indirect data dependence on $x_1$, denoted as $x_1 \rightsquigarrow_D z_3$.

While this example illustrates a simple chain of assignments, real-world programs often exhibit more complex forms of data dependence involving pointers, aliasing, and indirect memory access.

Accurately capturing such dependencies remains a key challenge for both traditional static analysis tools and LLM-based reasoning systems.

**Applications.** Data dependencies essentially induce the def-use chains in the program, serving as the ingredient for diverse static analysis applications, such as program slicing [24], compiler optimizations [42], and taint tracking in security [37, 43], with growing interest in using LLMs to assist such analyses [44, 45].

Fig. 1a presents an example of a real-world security vulnerability CVE-2017-5638 in Apache Struts2, which is caused by the usage of a program value that is data-dependent on an untrusted input. Here, $header_2$ receives untrusted input from an HTTP request and is used to define $expr_4$, forming a direct data dependency: $header_2 \rightarrow_D expr_4$. The value of $expr_4$ is then used at $\ell_6$ in a critical operation. This example demonstrates how data dependency analysis supports taint analysis by uncovering how untrusted input propagates through variable assignments to reach sensitive operations, making it a key technique for detecting and analyzing security vulnerabilities.

## 2.2 Control Dependency

Control dependency captures whether the execution of one statement is governed by another [46]. We use the line numbers of the statements to indicate the dependency. Specifically, a statement at $\ell_2$ is control-dependent on $\ell_1$ if the condition at $\ell_1$ determines whether $\ell_2$ executes. This holds when there is at least one branch from $\ell_1$ where $\ell_2$ always executes, and another where it may not.

```
1  if x > 0:        # controls execution of line 2
2      if y > 0:    # controls execution of line 3
3          a = 3    # only executes if both conditions hold
4  b = a + 1        # executes unconditionally
```

Here, $\ell_3$ is directly control dependent on $\ell_2$, and $\ell_2$ on $\ell_1$, forming the trace $\ell_1 \rightarrow_C \ell_2 \rightarrow_C \ell_3$. This trace represents a transitive control dependency: $\ell_1 \rightsquigarrow_C \ell_3$. $\ell_4$ is not control dependent on any condition in this code snippet. In our benchmark, tasks targeting control dependencies require LLMs to reason about how branching conditions affect execution paths. This involves identifying statements whose execution depends on earlier control conditions, including transitive cases across nested or compound conditionals.

**Applications.** Control dependencies capture the behavior of branches. They are widely used in path reachability reasoning in various program analysis tasks, such as bug detection [47, 48] and program verification [49, 50], which involves reasoning about the variables affecting the path execution.

Fig. 1b is an example of applying control dependency analysis in fuzzing, which presents a real-world vulnerability CVE-2022-26129 in the BABEL routing daemon. The call to memcpy at $\ell_{12}$ is the vulnerable operation that may cause a buffer overread. Reaching this line requires the input to satisfy a chain of conditions from $\ell_2$, $\ell_4$, $\ell_7$, $\ell_9$, and $\ell_{10}$. These conditionals form a transitive control dependency trace ending at $\ell_{12}$. Because triggering the vulnerable memory access depends on passing multiple branching checks, control dependency analysis is crucial for guiding fuzzers to explore such paths and uncover deep bugs [51, 52].

## 2.3 Information Flow

*Information flow* [53, 54] captures how the value of one variable can influence another through data or control dependencies. It may be *explicit*, as in direct assignments, or *implicit*, when control conditions determine which value a variable receives. An information flow may be a sequence of implicit, explicit, or mixed flows.

```
1  x = 0
2  if x == 0:
3      y = 0         # y is assigned if x is true, therefore a implicit flow from x
4  else:
5      y = 1         # alternative assignment to y
6  z = y + 1         # explicit flow from y
```

Here, the condition at $\ell_2$ directly reads $x_1$ and has direct control dependence over both $y_3$ and $y_5$, so $x_1 \rightarrow_I y_3$ (or $y_5$) is an implicit flow. $y_3 \rightarrow_I z_6$ (or $y_5 \rightarrow_I z_6$) is an explicit flow through assignment. Combining both gives a transitive information flow: $x_1 \rightsquigarrow_I z_6$. In our benchmark, tasks targeting

information flow require LLMs to reason over both explicit assignments and implicit control-induced flows, and to recover transitive chains of influence between variable definitions.

**Applications.** Information flow analysis is widely used in security and privacy, including taint tracking, non-interference, and enforcement of confidentiality and integrity policies [55]. In software engineering, it supports the program slicing and more downstream tasks, such as program debugging and fault localization. For example, it helps isolate the minimal subprogram relevant to a fault, improving code understanding and the developer's ability to identify and fix bugs [56, 57]. Recent research increasingly explores LLMs for assisting such analyses [58].

Fig. 1c is an example of slicing for debugging and fault localization, based on a real-world code snippet[3]: At $\ell_{13}$, the variable `in` may be null, leading to a potential NullPointerException (NPE). A static backward slice rooted at $\ell_{13}$ with respect to `in` reveals two possible assignments: $\ell_2$ and $\ell_{10}$. Whether these assignments execute depends on the condition at $\ell_1$, which govern the control flow paths. Thus, the slice includes both the relevant data dependencies and the control dependencies that determine whether `in` is properly initialized.

# 3 Our Benchmark: CORE

We introduce a human-verified high-quality benchmark for evaluating the semantic reasoning capabilities of LLMs in code. Unlike prior benchmarks targeting code generation or functional correctness, our objective is to provide a more fine-grained evaluation of LLMs' code semantic reasoning ability by investigating the three core semantic properties: **data dependency**, **control dependency**, and **information flow**. The benchmark is **multi-lingual** (C/C++, Java, and Python), and diverse in both task type and difficulty. It contains 12,553 tasks derived from 180 annotated programs, each manually curated or reviewed. A *semantics-aware sampling strategy* guides the construction of the task, ensuring both structural **diversity** and **reasoning complexity**. By emphasizing reasoning over program structure and behavior, this benchmark provides a targeted testbed to evaluate LLMs' understanding of the code semantics.

## 3.1 Benchmark Construction

We sample programs from two major sources: **CodeNet** [59], a large-scale dataset of competitive code, and **Google Code Jam (GCJ)** [60], which contains expert-written solutions to algorithmic problems. From these, we select 60 files each in C/C++, Java, and Python, resulting in 180 functions in total. We also select one target function from each file.

Our sampling ensures diverse program structures and a balanced distribution of lines of code (LoC) in each programming language. Specifically, we categorize candidate functions into four LoC buckets: 21–40, 41–60, 61–80, and 81–100, and ensure equal distribution among them. As function calls can introduce long dependency chains across functions, making it difficult to provide all the relevant functions in a single prompt, we only focus on *intra-procedural* analysis, excluding inter-procedural reasoning. We only select the functions that do not invoke any non-library functions, which is also a common practice in existing studies [24, 57], Check appendix A for benchmark construction details.

## 3.2 Data Annotation

For each function, we select a target variable and label its direct data and control dependencies. All variables transitively involved in these dependencies are also annotated. Due to the infeasibility of automating annotation (Appendix B.1), we adopt a semi-automated approach. A custom static analysis tool built on `tree-sitter` generates initial labels, which are then manually validated by two authors, each with over 5 years of program analysis experience. Conflicts are resolved by a third author. This process achieves an agreement rate of 87.5 %. In total, it yields 6,306 annotated variables and 48,050 lines of annotation. More annotation details are provided in the Appendix B.

---

[3]Originally from StackOverflow post #16180130.

Table 1: CORE task instance distribution

| Task Type | Postive | Negative | Total |
|---|---|---|---|
| Data Dependency | 1,814 | 2,800 | 4,614 |
| Control Dependency | 1,693 | 1,959 | 3,652 |
| Information Flow | 2,291 | 1,996 | 4,287 |
| Total | 5,798 | 6,755 | 12,553 |

Table 2: CORE Lite task instance distribution

| Task Type | Postive | Negative | Total |
|---|---|---|---|
| Data Dependency | 209 | 381 | 590 |
| Control Dependency | 239 | 250 | 489 |
| Information Flow | 291 | 214 | 505 |
| Total | 739 | 845 | 1,584 |

## 3.3 Task Design

We define three **task types** corresponding to core static analysis concepts: *data dependency*, *control dependency*, and *information flow* (Section 2). Each task type consists of multiple **task instances**. A task instance is a query that asks the model to analyze a specific type of dependency between program elements within a given program.

### 3.3.1 Task Formulation

While our raw annotations can support various static analysis tasks, we define and focus on two query types for each dependency category. Exact question formulations are provided in Appendix C.

**Pairwise Query.** Given two program elements (variables or lines), determine whether a specific dependency exists. If so, the model must also output a valid trace: a transitive sequence of direct dependencies or flows from source to target. When multiple traces exist, any one is sufficient.

**Target-Centric Query.** Given a single program element, list all other elements in the same function that have the specified dependency relation over it (e.g., all variables that the target is data dependent on). The model is expected to return a complete (orderless) set.

### 3.3.2 Task Instance Generation with Semantics-Aware Diverse Sampling

For each of the 180 annotated programs, we generate task instances for all three task types. For each task type, we sample up to five targets per program: variables for data dependency and information flow, and lines for control dependency. For each target, we construct up to five positive and five negative task instances. Each task instance poses a query about whether another variable or line in the same program holds a specific dependency relationship over the target.

We apply a *Semantics-Aware Diverse Sampling* strategy to guide both target and instance selection. The strategy is informed by code structure, such as control flow, as well as semantic dependencies to ensure both *diversity* and *reasoning complexity*. It promotes **diversity** by avoiding overlapping traces or repeated contexts, and enforces **reasoning complexity** by selecting targets with non-trivial dependency structures and negative instances that are structurally plausible to test the model's ability to distinguish true dependencies from misleading patterns.

This process results in a total of 12,553 task instances. Table 1 summarizes the distribution across task types and programming languages. Ground truth labels are automatically derived from the annotation. See Appendix D for sampling algorithms, statistics, and ablation study.

## 3.4 CORE Lite

To encourage broader adoption of CORE, we release a lite subset containing 1,584 task instances (Table 2). Based on the full benchmark constructed in Section 3.3.2, we randomly sample one instance per target, providing a smaller yet diverse and representative version. We evaluate the target-centric query only on CORE Lite, since each target appears only once, avoiding redundancy.

## 4 Experimental Setup

**Prompt Design.** Each prompt includes detailed definitions, the expected output format, and 5–7 illustrative examples drawn from small synthetic programs designed for demonstration (Appendix J). Also, each example includes step-by-step explanations and final outputs.

Table 3: Evaluated models with sizes and reasoning capabilities.

| | Claude 3.7 | Claude 3.5 | DeepSeek R1 | DeepSeek V3 | Gemini 2.5 Pro | GPT o3 | GPT o4-mini | GPT 4o | Llama3.1 | Qwen3 |
|---|---|---|---|---|---|---|---|---|---|---|
| Size | - | - | 671B | 671B | - | - | - | - | 405B | 235B |
| Reason | ✓ | ✗ | ✓ | ✗ | ✓ | ✓ | ✓ | ✗ | ✗ | ✓ |

Table 4: Model performance on various tasks on three task types: Data Dependency (Data), Control Dependency (Control), and Information Flow (InfoFlow) on CORE Lite.

| Models (Reasoning v.s Non-Reasoning) | Dependency Classification F1 (%) | | | | Trace Generation Correct Trace Rate (%) | | | | Dependency Source Enumeration Exact Match (%) | | | |
|---|---|---|---|---|---|---|---|---|---|---|---|---|
| | Data | Control | InfoFlow | Overall | Data | Control | InfoFlow | Overall | Data | Control | InfoFlow | Overall |
| Claude 3.7 | 76.94 | 88.84 | 81.57 | 82.07 | 69.29 | 60.04 | 39.90 | 57.13 | 20.58 | 51.70 | 6.92 | 25.82 |
| DeepSeek R1 | 83.29 | 92.28 | 83.59 | 86.18 | 67.31 | 66.62 | 39.58 | 58.36 | 38.88 | 48.37 | 7.12 | 31.82 |
| Gemini 2.5 Pro | 88.53 | **92.49** | **94.79** | 91.74 | **90.38** | **92.26** | **68.66** | **84.02** | **49.43** | **75.66** | **26.73** | **50.25** |
| GPT o3 | **93.24** | 92.11 | 92.13 | **92.56** | 86.23 | 77.52 | 52.37 | 72.80 | 41.89 | 70.90 | 15.61 | 42.61 |
| GPT o4-mini | 84.70 | 91.98 | 84.08 | 86.74 | 70.39 | 66.83 | 42.32 | 60.43 | 29.11 | 61.76 | 8.90 | 32.89 |
| Qwen3 235B | 82.33 | 89.00 | 69.51 | 80.31 | 65.48 | 59.19 | 29.85 | 52.26 | 21.36 | 44.61 | 4.74 | 23.30 |
| Claude 3.5 | 70.67 | 78.36 | 84.06 | 77.27 | 54.19 | 57.34 | 35.90 | 49.43 | 11.14 | 40.46 | 7.72 | 19.13 |
| DeepSeek V3 | 71.77 | 79.48 | 76.88 | 75.80 | 56.37 | 40.27 | 23.66 | 41.08 | 15.95 | 21.33 | 2.58 | 13.38 |
| GPT 4o | 69.75 | 77.11 | 76.63 | 74.16 | 61.85 | 43.15 | 22.53 | 43.56 | 12.28 | 29.06 | 2.97 | 14.52 |
| Llama 3.1 405B | 63.15 | 70.30 | 74.49 | 68.93 | 39.50 | 27.77 | 15.98 | 28.48 | 2.96 | 5.35 | 1.59 | 3.28 |

**Models.** Table 3 shows the 10 models we evaluate in this paper. Given the complexity and reasoning-oriented nature of our benchmark, we primarily focus on models with reasoning capabilities and strong coding performance, typically with sizes above 200B parameters. In total, we evaluate 6 reasoning and 4 non-reasoning models. Some non-reasoning models still show strong code performance despite lacking explicit reasoning abilities. Invocation and API details are in Appendix E.

**Evaluation Metrics.** We evaluate model performance across three aspects: *dependency classification*, *trace quality*, and *dependency source enumeration*.

- **Dependency Classification.** Model is queried whether a specific dependency relation holds between two program elements (variables or lines). We report precision, recall, and F1 score.
- **Trace Quality.** If a dependency is predicted to exist, the model must also produce a valid trace: a transitive sequence of direct dependencies or flows. We evaluate this using **Correct Trace Rate (CT)**, defined as the fraction of traces that are entirely correct.
- **Dependency Source Enumeration.** Given a target variable or line, the model is asked to list all elements (variables or lines) that influence it via a specific dependency type. We report **Exact Match (EM)**: the percentage of predictions that match the full ground-truth set.

Fine-grained metrics such as edge-level correctness and partial set recall are defined in Appendix F, where we also report complete results on those to support deeper analysis.

## 5 Evalution Results

We organize our evaluation around three questions: RQ1 (Section 5.1) reports overall performance, RQ2 (Section 5.2) presents a qualitative study of factors affecting model behavior, and RQ3 (Section 5.3) examines the impact of different experimental designs.

### 5.1 RQ1: How Well Do LLMs Perform on Dependency Reasoning Tasks?

Tables 4 report the performance of the 10 evaluated models on CORE Lite, split into reasoning (top) and non-reasoning (bottom) models. We evaluate dependency classification (whether a dependency exists), trace generation (producing a valid transitive trace), and dependency source enumeration (listing all sources for a given target) across data dependency, control dependency, and information flow. Full results on CORE with all metrics and more analysis are in Appendix G.

Reasoning models generally perform well on dependency classification, achieving over 80% F1 score. However, trace generation remains challenging: only Gemini 2.5 Pro and GPT-o3 achieve over 70% correct rate, while others lag at 20–60%. Dependency source enumeration appears to be the hardest, with most reasoning models performing below 40%, and non-reasoning models below 20%. Among all models, Gemini 2.5 Pro shows the best overall performance, followed by GPT-o3.

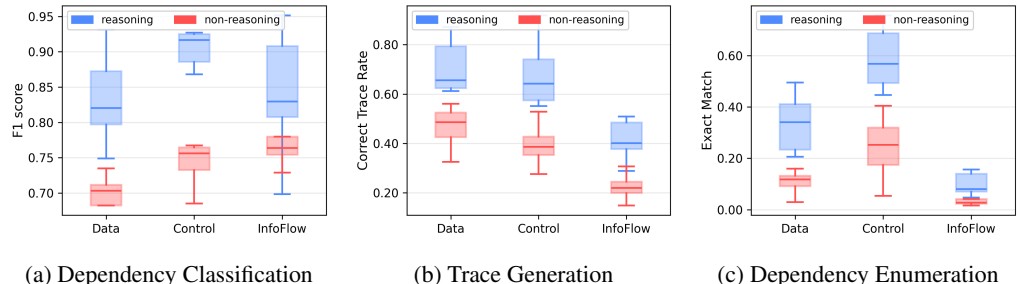

(a) Dependency Classification     (b) Trace Generation     (c) Dependency Enumeration

Figure 2: Performance distribution of reasoning vs. non-reasoning models across all three task types: Data Dependency (Data), Control Dependency (Control), and Information Flow (InfoFlow).

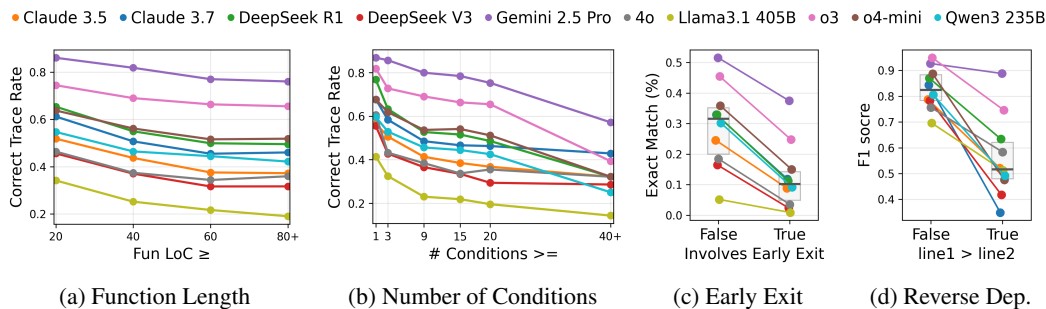

(a) Function Length     (b) Number of Conditions     (c) Early Exit     (d) Reverse Dep.

Figure 3: Factors affect model performance, with longer functions, complex control, and reverse dependencies leading to consistent drops.

Fig. 2 shows the performance distribution across all tasks. Reasoning models consistently outperform non-reasoning ones by a margin of 5.2–31.5%. Despite achieving strong classification F1 for information flow (Fig. 2a), models often struggle to produce correct traces for it (Fig. 2b), which requires integrating both explicit and implicit flows. A similar gap appears for control dependency, likely resulting from challenges in parsing control structures, especially under early returns or nested blocks (Section 5.2). Control dependency enumeration (Fig. 2c) is relatively easier due to its limited set of line-level sources (avg. 4.0), while data and information flow require enumerating more variable-level sources (avg. 7.7, 17.0), making them much more difficult.

As shown by the above statistics, while reasoning models can identify dependencies with high precision, generating accurate traces and enumerating all relevant sources remains a significant challenge, especially for information flow, which is the most challenging task as it requires reasoning over both data propagation and control influences.

## 5.2 RQ2: What Factors Influence Model Performance?

To better understand the challenges of the benchmark, we analyze factors that increase task difficulty and assess their impact on model performance.

**Function Length.** As shown in Fig. 3a, model performance (correct trace rate) on trace generation consistently declines as the length of the function increases. The performance gap between short (21–40 LoC) and long (81–100 LoC) functions reaches 8.9–15.8%. Longer functions present more tokens for the model to process, requiring it to reason over larger and often more entangled code contexts, including deeply nested logic and scattered dependencies.

**Control Structure Complexity.** We study how control structures affect trace generation quality. Fig. 3b shows the relationship between the number of conditions (e.g., `if`, `while`, `for`) in a function and the correct trace rate. As the number of conditions increases, model performance drops consistently, with differences of 24.4–44.6%.

We also examine the impact of early exits such as `return` or `break`. In Fig. 3c, we compare the exact match rate of dependency source enumeration between cases where the target variable is influenced by early exit logic and those where it is not. Across all models, we observe a consistent drop in performance of up to 21.7%.

These findings suggest that complex or non-linear control flows significantly increase reasoning difficulty, likely due to the need to track multiple execution paths and conditionally gated behaviors.

**Reverse Dependency Order.** With the pair-wise query, each query involves two program elements (variables or lines), such as "Does variable $a$ have a data dependency over variable $b$?" Typically, the source element ($a$) appears earlier in the code than the target ($b$), matching natural expectations (i.e., $\texttt{line}(a) < \texttt{line}(b)$). However, reverse dependencies also occur, especially in loops, where a later variable can influence an earlier one. Fig. 3d shows how this affects classification F1 scores. When the source appears after the target, model performance drops significantly, by up to 49.5% (Claude 3.7). This suggests that LLMs are biased toward left-to-right reasoning and struggle with abstract dependency graphs that require backward or bidirectional analysis.

### 5.3 RQ3: How Do Different Experimental Settings Affect Model Behavior?

We evaluate the impact of three experimental variations on model performance, focusing on how task framing and input context influence model behavior.

**Asking for Traces vs. Classification Only.** In the default setting with pair-wise query, models are asked to predict whether a dependency exists and provide a supporting trace. We compare this to a simplified classification-only setup, where no trace is requested. As shown in Fig. 4a, this results in higher precision but lower recall. This suggests that when prompted to produce a trace, models are more likely to answer "yes" (i.e., assert that the relationship holds).

**Clipping the Function Context.** By default, the entire file is provided as input, and the prompt specifies the target function. We compare this to a clipped setup that includes only the target function. Fig. 4b shows that models perform consistently better in the clipped setting, improving by up to 5.7%, indicating that models struggle to focus when given longer irrelevant context, even with a specified function, in contrast to traditional analysis tools, which inherently isolate scope.

**Few-Shot Learning (FSL).** While our base prompt includes examples and definitions, we further explore an FSL setting using additional examples retrieved via sparse and dense retrievers. Surprisingly, results do not improve and even degrade, likely due to (1) longer inputs distracting from core instructions, and (2) potential bias from retrieved examples, even with balanced labels. Additionally, standard retrievers fail to capture structural or semantic similarity between tasks, suggesting that better structure-aware retrieval is needed. Details are presented in Appendix H.

## 6 Related Work

### 6.1 LLMs for Software Engineering Tasks

LLMs have been widely applied across many domains [61, 62, 63, 64], including software engineering, where they are used for tasks such as programrepair [10, 11, 12, 65], code generation [15, 20, 66, 67, 68], software testing [13, 69, 70, 71, 72], and vulnerability detection [17, 45, 73]. A key enabler for these applications is reasoning over static program properties. Traditional static analysis tools, however, are language-specific and require extensive manual engineering. To overcome these limitations, recent works increasingly use LLMs themselves to perform static analysis as an intermediate step. LLMDFA [44] and E&V [74] apply LLM-based data- and control-flow reasoning to detect bugs; IRIS [45] leverages LLMs to examine data-flow paths reported by static analyzers, effectively mitigating the false positives. CORE offers a fine-grained benchmark to evaluate LLMs' capabilities in reasoning about program semantics, which underlie these downstream SE tasks.

### 6.2 Code Reasoning Benchmark and Evaluation

Code reasoning, the ability to understand program behavior, is fundamental to downstream LLM applications such as program synthesis [75, 76, 77], program repair [22, 78, 79], and bug or vulnerability detection [80, 81, 17, 18]. These tasks require models to reason over program semantics,

including data and control flow. Yet, most benchmarks only measure downstream success (e.g., detecting vulnerabilities successfully) without directly probing models' reasoning capabilities on code semantics.

Recent work has begun to explicitly evaluate internal code reasoning. CRUXEval [82] focuses solely on Python and evaluates input-output behavior, which aligns more with dynamic analysis. In contrast, our benchmark targets static analysis skills and covers Python, Java, and C/C++. CRQBench [83] evaluates LLMs' semantic reasoning with C++ questions focused on variable tracking and behavioral equivalence. However, its questions focus on high-level behaviors and are automatically generated from pull requests and review comments, making them error-prone. Similarly, [67] evaluates finetuned LLMs on several program analysis tasks, such as memory region prediction and function signature recovery from Java programs. However, the tasks remain high-level. CodeMMLU [21] focuses on high level also focuses on knowledge-based high-level questions. REval [84] focuses on programs' runtime behaviors such as code coverage prediction. CodeCrash [85] evaluates model robustness via adversarial code perturbations, but does not directly assess semantic reasoning. CORE fills this gap by enabling fine-grained assessment of models' fundamental code understanding capabilities, including data and control dependencies and information flow, beyond surface-level pattern recognition or input-output matching.

## 7 Conclusion

We introduce CORE, a high-quality multilingual benchmark for evaluating LLMs on fundamental static analysis tasks, including data dependency, control dependency, and information flow across C/C++, Java, and Python. It includes 12,553 human-verified, diverse instances from 180 programs. Our evaluation of 10 state-of-the-art LLMs shows that while many models perform well at identifying dependencies, they struggle with deeper semantic tasks such as generating valid traces. Our qualitative analysis further highlights key challenges such as long functions, complex control structures, and backward reasoning. CORE fills an important gap by enabling fine-grained assessment of fundamental program reasoning skills beyond surface-level cues.

**Future Directions.** A natural extension is to support inter-procedural analysis. Despite the difficulty of obtaining reliable annotations, one possible approach is to employ lightweight static analysis tools to identify function pairs with dependencies (e.g., via calling context). Once each function is fully annotated, a call graph can be constructed to enable inter-procedural reasoning.

Another promising direction is to examine the resilience of LLMs' code reasoning under various forms of code obfuscation or subtle adversarial perturbations. Such analysis could shed light on the robustness and reliability of their semantic understanding in complex real-world scenarios.

## 8 Limitations

Our benchmark focuses on relatively short (within 100 LoC) due to the difficulty of scaling automated construction (Section B) and significant manual effort in annotation (Section 3.2). However, the functions within this size range still reflect real-world practices, as well-structured codebases often keep functions concise for maintainability (e.g., 40–50 LoC) [86, 87]. Nonetheless, we curate 12,553 instances with an assurance of diversity and difficulty (Section 3.3.2). While annotation is manual and may introduce errors, we alleviate this through sufficient cross-validation (Section 3.2).

## 9 Acknowledgement

We are grateful to the Center for AI Safety for providing computational resources. This work was funded in part by the National Science Foundation (NSF) Awards 2006688, 1901242, SHF-1901242, SHF-1910300, Proto-OKN 2333736, IIS-2416835, DARPA VSPELLS - HR001120S0058, ONR N00014-23-1-2081, and Amazon. Any opinions, findings and conclusions or recommendations expressed in this material are those of the authors and do not necessarily reflect the views of the sponsors.

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

Table 5: Distribution of sampled functions by dataset, language, and function length (in LOC).

| Dataset Source | Lang. | Target Function LoC | | | | Total |
|---|---|---|---|---|---|---|
| | | 21–40 | 41–60 | 61–80 | 81–100 | |
| | C/C++ | 7 | 7 | 6 | 10 | 30 |
| CodeNet | Java | 8 | 7 | 8 | 7 | 30 |
| | Python | 7 | 6 | 7 | 10 | 30 |
| Google Code Jam | C/C++ | 8 | 8 | 9 | 5 | 30 |
| (GCJ) | Java | 7 | 8 | 7 | 8 | 30 |
| | Python | 8 | 9 | 8 | 5 | 30 |
| Total | | 30 | 30 | 30 | 30 | 180 |

# A  Benchmark Construction

## A.1  Dataset Sources

**CodeNet.**  CodeNet [59] is a large-scale dataset introduced by IBM. It consists of over 14 million code submissions spanning 55 programming languages, with a total of approximately 500 million lines of code. The dataset is derived from competitive programming problems and includes rich annotations such as problem IDs, programming language labels, code correctness (based on test case results), runtime and memory usage, and input-output test cases for the majority of problems. These annotations make CodeNet particularly valuable for supervised learning tasks and for constructing benchmarks focused on correctness, efficiency, or cross-language analysis. In our work, we use CodeNet as a source of diverse, well-labeled, real-world functions that reflect authentic coding practices across a variety of problem types and difficulty levels. The dataset is licensed under Apache 2.0.

**Google Code Jam (GCJ).**  Google Code Jam [60] is a long-running global programming competition hosted by Google, which features algorithmic problems of varying complexity. GCJ problems are typically accompanied by official test sets and are designed to test a range of skills including algorithm design, edge-case handling, and optimization under constraints. Solutions submitted by participants are generally concise, high-quality, and written under time pressure, providing a rich source of clean, focused code samples. For our benchmark, we extract complete solution files from GCJ archives, which offer naturally diverse control and data flow patterns that are especially well-suited for program analysis tasks. Specifically, we only sample the code from 2018 – 2021 collected from a public GitHub repository [4] that crawled data from the GCJ website. The repository is with an MIT license.

## A.2  Data Sampling

Table 5 shows the distribution of sampled functions by dataset, language, and function length. We sample 180 program files in total, 90 from CodeNet and 90 from Google Code Jam (GCJ), and select one target function from each file for annotation.

Among the remaining files, we first remove all the comments and empty lines, and categorize candidate functions into four LoC buckets: 21–40, 41–60, 61–80, and 81–100. We enforce an even distribution across all buckets and across the three target languages (C/C++, Java, Python), resulting in a balanced and diverse set of functions suitable for dependency analysis.

To ensure consistent analysis scope, we discard samples that involve inter-procedural constructs such as function calls with non-local context or reliance on global variables. We also exclude files longer than 300 lines of code to maintain manageable complexity.

When necessary, we apply minimal, semantics-preserving edits to facilitate annotation. For example, we expand inline expressions or break compound statements (e.g., multiple statements on the same line separated by ";" in C or Java) into separate lines. These edits help standardize the structure of the code while preserving its original behavior.

---

[4] https://github.com/Jur1cek/gcj-dataset

# B   Data Annotation

## B.1   Challenges for Automation

As discussed in Section 2, the program properties addressed by our reasoning tasks are fundamentally semantic properties. According to Rice's Theorem [88], determining whether a semantic property holds or not is inherently undecidable. Consequently, it is infeasible to develop an algorithm that fully automates the data annotation process and yields the prefect ground truth of the targeted semantic properties. Initially, we also explored employing static and dynamic analysis techniques to automatically generate data and control dependencies. However, we discovered that existing techniques are insufficient to significantly alleviate the annotation overhead.

**Existing Static Analyzers.**   Existing static analyzers, such as SVF [89], FlowDroid [43], CodeQL [90], and KLEE [91], predominantly analyze intermediate representation (IR) code rather than source code directly. IR code, which is generated by compiler frameworks, facilitates foundational analyses like pointer analysis [92]. Hence, the outputs of these analyzers typically depict the relations among program values within IR code rather than the values of source-level variables and expressions. However, LLMs operate directly on source code, not IRs, creating a semantic gap between traditional analysis tools and the kind of source-level reasoning we aim to evaluate. As a result, existing static analysis tools cannot be directly used to generate the labels required for our benchmark.

Furthermore, existing static analyzers are primarily designed for languages like C/C++ and Java, leaving Python, widely prevalent in the machine learning community, largely unaddressed. To the best of our knowledge, no practical Python static analyzer can accurately derive the specific semantic properties targeted by this work. Hence, utilizing static analyzers to automate data annotation is currently impractical.

**Existing Dynamic Analyzers.**   Dynamic analysis [93, 94, 95], unlike static analysis, is generally more language-agnostic. For given inputs, it readily tracks variable and expression values throughout program execution as long as we have the execution environment. Nevertheless, dynamic analysis only captures program behavior corresponding to specific inputs, potentially leaving many program behaviors unexplored. As underscored by prior dynamic testing research [96, 97], achieving sufficient coverage to comprehensively characterize all program behaviors is challenging. Despite advancements such as fuzz testing aimed at increasing coverage [98], complex control-flow structures, such as branches and loops, still significantly restrict the completeness of collected data and control dependencies.

Considering these limitations, we abstain from employing conventional static or dynamic analysis techniques for automating data annotation. Instead, we introduce the methodology detailed in Section 3, designed specifically to generate diverse, high-quality datasets applicable across multiple programming languages.

## B.2   Annotation Rules

For each annotated data point (i.e., a sampled function), we randomly select one target variable. We then annotate all its **direct** data and control dependencies, and recursively expand the annotation to include all transitive dependencies. This ensures that complete dependency traces can be constructed during evaluation. We only annotate data and control dependencies, as information flow can later be derived from these annotations. The annotation strictly follows the definitions introduced in Section  2.1 and 2.2.

An example of a sampled C function from CodeNet [5] and its corresponding annotation is shown below.

---

[5]The sampled function `main`, spanning lines 12–40, is from CodeNet problem `p00496`, solution `s700056700`.

```
10 | ...
11 |
12 | int main()
13 | {
14 |   int a[3001];
15 |   int b[3001];
16 |   int n, t, s, i, j, k, ans;
17 |   fgets(p=buf, 30, stdin);
18 |   n = getint();
19 |   p++;
20 |   t = getint();
21 |   p++;
22 |   s = getint();
23 |   for (i = 1; i <= n; i++) {
24 |     fgets(p=buf, 30, stdin);
25 |     a[i] = getint();
26 |     p++;
27 |     b[i] = getint();
28 |   }
29 |   ans = 0;
30 |   for (i = 1; i <= n; i++)
31 |     for (j = 1; j <= t; j++) {
32 |       dp[i][j] = max(dp[i-1][j], dp[i][j-1]);
33 |       k = j-b[i];    // <- The target variable
34 |       if (k >= 0 && (s <= k || j <= s))
35 |         dp[i][j] = max(dp[i][j], dp[i-1][k] + a[i]);
36 |         ans = max(ans, dp[i][j]);
37 |     }
38 |   printf("%d\n", ans);
39 |   return 0;
40 | }
```

```
 1 | target_var: k 33
 2 | other_vars:
 3 |   - a 25
 4 |   - p 26
 5 |   - dp 32
 6 |   - dp 35
 7 | variables:
 8 |   k 33:
 9 |     datadep:
10 |       - j 31
11 |       - b 27
12 |       - i 30
13 |     controldep:
14 |       - j 31
15 |       - t 31
16 |   b 27:
17 |     datadep:
18 |       - i 23
19 |     controldep:
20 |       - i 23
21 |       - n 23
22 |   t 31:
23 |     reset: False
24 |     datadep:
25 |       - t 20
26 |     controldep:
27 |       - i 30
28 |       - n 30
29 |       - j 31
30 |       - t 31
31 |   ...
```

Listing 1: Source code      Listing 2: Annotation

We annotate control dependencies at the **variable level** rather than the line level, as variable-level granularity is necessary for deriving information flow. Variables are represented in the (`name, line`) format (similar to the format introduced in Section 2), referring to where the variable is (re)defined or updated. We do not annotate variables that are only read at a line (e.g., y used in x = y + 1 without being updated).

Annotations are stored in a human-readable YAML format to facilitate manual editing and automated parsing. The YAML files consist of:

- `target_var`: the selected target variable. Here it is (`k,33`), which is variable k defined in line 33 in the source code.
- `other_vars`: variables annotated as irrelevant. In addition to annotating all variables involved in direct or transitive dependencies, annotators also labeled a set of *irrelevant variables* to serve as negative examples. These include variables that appear before the target line or within the same loop but are NOT semantically dependent on the target. These serve as negative examples to test a model's ability to distinguish true semantic influence from structural proximity.
  For example, the variable a defined at line 25, (`a,25`), has no data or control dependencies leading to the target variable (`k,33`). That is, there is no information flow from (`a,25`) to (`k,33`). As a result, (`a,25`) is labeled as an `other_var`, as shown on line 3 in the annotation.
- `variables`: each variable annotated with its direct data dependencies (`datadep`) and control dependencies (`controldep`).

Note that while we only annotate variables at the lines where they are (re)defined, an exception arises when assigning control dependencies. Specifically, the control-dependent lines for a variable may not themselves redefine that variable. For example, variable (`k,33`) is directly control-dependent on line 31. Therefore, it is annotated with control dependencies (`j,31`) and (`t,31`), even though t is not assigned or updated at line 31. To represent this dependency consistently using our (`name, line`) format, we include such cases in the annotation but mark them with "`reset:False`" (line 23 in the

annotation). This flag ensures that these entries are excluded from data dependency processing while still being used for control and information flow reasoning.

## B.3   Annotation Statistics

On average, each function contains 35.03 annotated variables as participating in a dependency or information flow relation. Each annotated variable has 1.92 direct data dependencies and 1.92 direct control dependencies.

The average manual annotation time is 37 minutes per annotator per function. To ensure annotation quality, each data point is cross-verified by at least two authors. Disagreements are resolved by a third author, with an observed agreement rate of 87.5 %.

## C   Task Formulation

We list the query templates used for both pairwise dependency and dependency source enumeration queries across data dependency, control dependency, and information flow, with the task-specific content (e.g., code snippets, variable names) highlighted in blue.

Note that these are only the query templates. For the full prompt templates used in our experiments, including task definitions, instructions, and examples, please refer to Appendix J.

---

Query for Pairwise Dependency Query (Data Dependency)

Below is **your target code snippet**.

`<target code with line number>`

**Question**:   Does `(src_var, src_line)` have data dependence over `(dst_var, dst_line)` in function `target_function_name`? If so, provide a trace.
**Output**:

---

Query for Dependency Source Enumeration (Data Dependency)

Below is **your target code snippet**.

`<target code with line number>`

**Question**: Which variable instances have data dependence over `(dst_var, dst_line)` in function `target_function_name`? List all such variables.
**Output**:

---

Query for Pairwise Dependency Query (Control Dependency)

Below is **your target code snippet**.

`<target code with line number>`

**Question**:   Does `src_line` have control dependence over `dst_line` in function `target_function_name`?   If so, provide a trace.
**Output**:

---

Query for Dependency Source Enumeration (Control Dependency)

Below is **your target code snippet**.

`<target code with line number>`

**Question**:   Which lines have control dependence over `dst_line` in function `target_function_name`?   List all such lines.

---

Query for Pairwise Dependency Query (Information Flow)

Below is **your target code snippet**.

`<target code with line number>`

**Question**:   Is there information flow from `(src_var, src_line)` to `(dst_var, dst_line)` in function `target_function_name`? If so, provide a trace.
**Output**:

# D  Semantics-Aware Diverse Sampling

For each of the 180 annotated programs, we generate task instances for all three task types. For each type, up to five target variables (or lines) are selected per program. For each target, we sample up to five positive (if the specified dependency relationship holds) and five negative examples (otherwise) with **enforced diversity**, resulting in 12,553 task instances in total.

To construct our benchmark, we adopt a *Semantics-Aware Diverse Sampling* strategy that governs both target selection and task instance generation. The following sections describe how this sampling strategy is applied to select targets (Section D.1) and generate positive/negative task instances (Section D.2) for each task type.

## D.1  Target Sampling

To construct a diverse and challenging benchmark, we sample up to five target variables (or lines) per program for each task type. A target serves as the anchor for generating task instances: positive and negative queries related to its dependencies. The sampling strategies are designed to enforce both *diversity*, by avoiding overlapping dependency traces, and *complexity*, by preferring targets involved in deeper or more nuanced dependency structures. While data dependency and information flow targets are sampled at the variable level based on transitive relationships, control dependency targets are line-based. Below, we describe the sampling procedures specific to each task type.

### D.1.1  Data Dependency Target Sampling

Algorithm 1 describes how we sample target variables for data dependency tasks. This algorithm is used only for selecting target variables; positive and negative pairs are sampled separately and will be introduced later. Our goal is to ensure both *diversity* and *complexity* in the selected targets.

In function `SampleDataDepTargets`, given a program and an annotated initial target variable $v_t$, we aim to select up to $n = 5$ target variables. The set $T$ is initialized with $v_t$ (line 2), and we maintain a set $S$ containing all transitive data-dependency sources of $v_t$ (Line 3). This helps avoid overlapping or redundant sampling. The function `GetAllDataDepSources` returns all variables that are transitively or directly data-dependent sources of the input variable.

Then, for each variable $v$ in the program (after random shuffling), we iterate through candidates and skip any that: (a) have already been selected, (b) appear in the transitive source set $S$ (i.e., are reachable from already-selected variables), (c) are not associated with a value assignment (i.e., `reset` is `False`), or (d) fail the structural filtering criteria defined in `IsValidDataDepTarget` (Line 9). In particular, if a variable $v$ is already in $S$, we exclude it because its data dependency sources would form a strict subset of those already covered. Any dependency trace from these sources to $v$ would also significantly overlap with traces of previously selected variables, reducing overall diversity.

We call `IsValidDataDepTarget` (lines 16–32) to ensure that the variable exhibits a non-trivial dependency structure. The function checks whether there exists a data-dependency source variable $v_s$ such that a transitive data dependency trace from $v_s$ to the input variable $v$ satisfies either of the following conditions:

(a) The trace contains at least two edges (i.e., has a dependency depth $\geq 2$), or

(b) It involves at least one unique upstream variable name (excluding $v.name$ itself).

The function uses a DFS-style traversal initialized with a stack. Each element of the stack is a tuple containing three elements:

- The current variable

**Algorithm 1:** Sample Data Dependency Task Instances

---

**1 Function** `SampleDataDepTargets`($F$, $n$, $v_t$)**:**
    **Input:** Function $F$, sample size $n$, initial target variable $v_t$
    **Output:** Set $T$ of sampled target variables
**2**     $T \leftarrow \{v_t\}$ ;                    `// Start with annotated target`
**3**     $S \leftarrow$ `GetAllDataDepSources`($v_t$) ;    `// Track seen sources to reduce`
     `duplication`
**4**     Shuffle all variables $V$ in $F$;
**5**     **foreach** $v \in V$ **do**
**6**         **if** $|T| \geq n$ **then**
**7**             | **break**
**8**         **end**
**9**         **if** $v \in T$ **or** $v \in S$ **or** $v$.reset $= False$ **or** $\neg IsValidDataDepTarget(v)$ **then**
**10**           | **continue**
**11**         **end**
**12**         $T \leftarrow T \cup \{v\}$;
**13**         $S \leftarrow S \cup$ `GetAllDataDepSources`($v$);
**14**     **end**
**15**     **return** $T$

**16 Function** `IsValidDataDepTarget`($v$)**:**
**17**     $stack \leftarrow [(v, 0, \emptyset)]$;
**18**     **while** *stack not empty* **do**
**19**         $(curr, dist, visited\_sources) \leftarrow stack.pop()$;
**20**         **foreach** $src \in curr.data\_dep$ **do**
**21**             **if** $src \in visited\_sources$ **then**
**22**               | **continue** ;               `// Skip already visited`
**23**             **end**
**24**             $d \leftarrow dist + 1$;
**25**             $S \leftarrow visited\_sources \cup \{src.name\}$;
**26**             **if** $d \geq 2$ **or** $|S| \geq 1$ **then**
**27**               | **return** *True*
**28**             **end**
**29**             Push $(src, d, S)$ to stack;
**30**         **end**
**31**     **end**
**32**     **return** *False*

---

- Its distance from the input variable. Here, distance refers to the number of direct data dependency edges between the current variable and v, i.e., the length of the current transitiva data dependency trace. This value is used to determine whether the variable lies on a sufficiently deep trace (e.g., two or more edges), which is one of the criteria for accepting a variable as valid.

- A set of source variable names already visited along the current path. The per-path tracking helps avoid revisiting nodes and prevents cycles during the depth-first traversal.

In each iteration (lines 18–31), we pop one tuple, and iterate over the **direct** data dependency sources of the current variable (line 20). We skip already visited sources to avoid cycles (lines 21–23). If the chain meets the success condition (line 26), we return `True`; otherwise, we continue exploring.

This approach ensures that each selected variable induces a trace that is sufficiently distinct and semantically meaningful.

### D.1.2   Control Dependency Target Sampling

The control dependency sampling strategy (Algorithm 2 `SampleControlDepTargets`) follows the same template as data dependency target sampling but introduces line-based filtering to enforce structural diversity across control paths.

**Algorithm 2:** Sample Control Dependency Target Variables

---

**1 Function** SampleControlDepTargets($F$, $n$, $v_0$, $depth$):
 **Input:** Function $F$, sample size $n$, annotated target variable $v_0$, max depth $depth$
 **Output:** Set $T$ of selected target lines
**2** $T \leftarrow \{v_0.line\}$;
**3** $L \leftarrow$ GetControlDepLines$(v_0, depth)$ ;      // Blocked control-dep lines
**4** Shuffle all variables $V$ in $F$;
**5** **foreach** $v \in V$ **do**
**6**  **if** $|T| \geq n$ **then**
**7**   **break**
**8**  **if** $v.line \in T$ **or** $\neg IsValidControlDepTarget(v)$ **then**
**9**   **continue**
**10**  $C \leftarrow$ GetControlDepLines$(v, depth)$;
**11**  **if** $L \cap C \neq \emptyset$ **then**
**12**   **continue** ;       // Avoid overlapping control paths
**13**  $T \leftarrow T \cup \{v.line\}$;
**14**  $L \leftarrow L \cup C$;
**15** **return** $T$

**16 Function** IsValidControlDepTarget($v$):
**17** stack $\leftarrow [(v, 0)]$;
**18** visited $\leftarrow \{v.\text{line}\}$;
**19** **while** *stack not empty* **do**
**20**  $(curr, dist) \leftarrow$ stack.pop();
**21**  **foreach** $p \in curr.\text{control\_dep}$ **do**
**22**   **if** $p.\text{line} = curr.\text{line}$ **or** $p.\text{line} \in visited$ **then**
**23**    **continue**
**24**   $d \leftarrow dist + 1$;
**25**   **if** $d \geq 2$ **then**
**26**    **return** *True*
**27**   visited $\leftarrow$ visited $\cup \{p.\text{line}\}$;
**28**   Push $(p, d)$ to stack;
**29** **return** *False*

---

The key difference is that instead of tracking dependency sources, we track the set of control-dependency *lines* visited so far. Specifically, for each candidate variable $v$, we use GetControlDepLines to extract the line numbers associated with its control dependencies, up to a specified depth $depth$. This is necessary because, in real-world code, certain lines, such as early return statements or large enclosing while loops, often dominate the control flow of the entire function. Without filtering, many variables would end up sharing identical or heavily overlapping control dependency traces, effectively blocking most of the lines before the loop even starts. By restricting overlap only within the first few levels of control dependency, we avoid such global blocking and enable the selection of variables that differ in their *local* control context, thus improving structural diversity in the sampled targets. If any of these lines overlap with those already visited, the candidate is skipped to avoid redundancy in control context (lines 11–12). The depth is set to 2.

The function IsValidControlDepTarget (lines 16–29) ensures that the variable is transitively control-dependent on at least two other lines (i.e., a path of length $\geq 2$). This is checked via depth-first traversal over control-dependency edges. Like before, only variables with meaningful and non-trivial dependency structure are retained.

### D.1.3 Information Flow Target Sampling

The sampling algorithm for information flow targets (Algorithm 3) extends the structure used in data dependency sampling (Section D.1.1 but differs in several key ways.

**Algorithm 3:** Sample Information Flow Target Variables

---

1 **Function** SampleInfoFlowTargets($F$, $n$, $v_t$):
  **Input:** Function $F$, sample size $n$, annotated target variable $v_t$
  **Output:** Set $T$ of selected target variables
2   $T \leftarrow \emptyset$;
3   $S \leftarrow \emptyset$;       // Track seen source variables across candidates
4   Shuffle all variables $V$ in $F$;
5   **foreach** $v \in V$ **do**
6     **if** $|T| \geq n - 1$ **then**
7       **break**
8     **if** $v \in T$ **or** $v \in S$ **or** $v$.reset $=$ *False* **or** $\neg IsValidInfoFlowTarget(v)$ **then**
9       **continue**
10     $U \leftarrow$ GetAllInfoFlowSources$(v, d)$;
11     $T \leftarrow T \cup \{v\}$;
12     $S \leftarrow S \cup U$;
13   $T \leftarrow T \cup \{v_t\}$;       // Add target after sampling
14   **return** $T$

15 **Function** IsValidInfoFlowTarget$(v)$:
16   **if** $IsValidDataDepTarget(v) =$ *False* **then**
17     **return** *False*
18   $D \leftarrow$ GetAllDataDepSources$(v)$;
19   $D \leftarrow D \cup \{v\}$;
20   **foreach** $x \in D$ **do**
21     **if** $x$.control_dep $\neq \emptyset$ **then**
22       **return** *True*
23   **return** *False*

---

First, we do not initialize the selected set with the annotated target variable $v_t$ (line 2). Doing so would cause the seen source set $S$ to immediately block all annotated variables (see annotation process in Section 3.2). Instead, we defer adding $v_t$ until after sampling (line 13).

Second, the validity check is stricter. The function `IsValidInfoFlowTarget` (lines 15–23) requires that the variable meet the same structural criteria as in the data dependency case (line 16), *and* that at least one of its data-dependency sources (including itself) is control-dependent on at least one variable. This ensures that selected targets participate in implicit information flows.

Finally, the dependency sources are gathered using `GetAllInfoFlowSources` (line 10), which aggregates all upstream control and data dependencies. Aside from these differences, the traversal, blocking, and shuffling logic mirror those used in data dependency target sampling.

### D.2 Positive and Negative Task Instances Sampling

For each selected target (a variable or line, depending on the task type), we construct up to five positive and five negative task instances to evaluate the model's ability to identify program dependencies. Each instance consists of a query: whether a specified dependency relationship holds between two program elements, together with a ground-truth answer (`boolean`) with a trace (if the relationship holds). While the overall sampling strategy is consistent across task types, the specifics vary based on the dependency semantics: data dependency and information flow tasks operate at the variable level, while control dependency operates at the line level and is restricted to syntactic condition constructs. We use prioritized sampling strategies to ensure negative examples are both structurally plausible and semantically incorrect (i.e., the relationship does not hold), thereby maximizing evaluation difficulty. Below, we detail the sampling procedure for each task type.

**Algorithm 4:** Sample Negative Task Instances for Data Dependency

---

**1 Function** SampleNegativeDataDepInstances($v_t$, $V$, $OtherVars$):

  **Input:** Target variable $v_t$, set of all variables $V$, set of irrelevant variables $OtherVars$
  **Output:** Two sets of variables: primary and secondary candidates

**2**  $D \leftarrow$ GetAllDataDepSources($v_t$);

**3**  $P \leftarrow \emptyset$ ;                    // Primary candidates: before target

**4**  $S \leftarrow \emptyset$ ;                   // Secondary candidates: loop-local after target

**5**  **foreach** $x \in V \cup$ OtherVars **do**

**6**    **if** $x == v_t$ **or** $x \in D$ **or** $x$.reset $= False$ **then**

**7**      **continue**

**8**    **if** $x$.line $< v_t$.line **then**

**9**      $P \leftarrow P \cup \{x\}$;

**10**    **else if** InSameLoopBlock($v_t$.line, $x$.line) **then**

**11**      $S \leftarrow S \cup \{x\}$;

**12**  **return** $P, S$

---

### D.2.1 Sampling Task Instances for Data Dependency

Given a selected target variable $v_t$, we sample up to five **positive** and five **negative** task instances for data dependency. Each instance is a variable that can form a query of the form "Does variable $x$ have data dependency over $v_t$?", along with a ground-truth label and, if applicable, a valid dependency trace.

**Positive Sampling.** Positive examples are sampled directly from the transitive data dependency sources of $v$ using GetAllDataDepSources (Algorithm 1 line 3). These variables are, by definition, those on which $v_t$ is data-dependent (either directly or transitively).

**Negative Sampling.** Negative examples are chosen among variables that do *not* have any transitive or direct data dependency over $v_t$, but are structurally close enough to be potentially confusing.

We use the strategy shown in Algorithm 4 to identify two prioritized pools of negative candidates:

- **Primary Candidates** ($P$): variables that appear on control lines **before** $v_t$ and do not have data dependency over $v_t$. These are likely to be mistaken as influences from upstream data.

- **Secondary Candidates** ($S$): variables that appear **after** $v_t$ but are in the same loop body (line 10). This accounts for reverse-order influence that often arises in loops, while still ensuring no true data dependency exists.

We sample up to five negative instances, prioritizing from $P$ and using $S$ only when the primary pool is insufficient. This prioritized sampling helps create hard negatives that challenge models to distinguish structural correlation from semantics.

### D.2.2 Sampling Task Instances for Control Dependency

For control dependency, each task instance consists of a program and a pair of line numbers, forming a question of the form: "Does line $x$ have control dependency over line $y$?". The model must answer this question along with a supporting trace if the answer is positive. Both positive and negative candidates are sampled from *condition lines only*, such as those introduced by if, while, or for statements.

**Positive Sampling.** Positive control dependency instances are obtained using GetControlDepLines($v_t$) (Algorithm 2 line 3), which returns all condition lines that have direct or transitive control over the target line $v_t.line$. These are directly extracted from the control dependency annotations and require the model to understand whether the condition influences the execution of $v_t$.

**Algorithm 5:** Sample Negative Task Instances for Control Dependency

---

**1 Function** SampleNegativeControlDepInstances($v_t$):

    **Input:** Target variable $v_t$

    **Output:** Two lists of candidate condition lines: primary and secondary

**2**     $C \leftarrow$ GetConditionBlocks() ;     `// All condition blocks of the function`

**3**     $D \leftarrow$ GetControlDepLines($v_t$);

**4**     $P \leftarrow \emptyset$ ;     `// Primary: non-controlling lines before target`

**5**     $S \leftarrow \emptyset$ ; `// Secondary: non-controlling lines after target in same loop`

**6**     **foreach** $cond\_line \in C$ **do**

**7**         **if** $cond\_line \in D$ **then**

**8**             **continue**

**9**         **if** $cond\_line < v_t$.line **then**

**10**             $P \leftarrow P \cup \{cond\_line\}$;

**11**         **else if** $cond\_line > v_t$.line **and** InSameLoopBlock($v_t$.line, $cond\_line$) **then**

**12**             $S \leftarrow S \cup \{cond\_line\}$;

**13**     **return** $P, S$

---

**Negative Sampling.** Negative candidates are sampled using Algorithm 5. Unlike arbitrary lines, we restrict negative sampling to lines that are syntactic conditions in the source code, ensuring that each candidate is a plausible controller. We extract all such condition blocks within a program (represented as $C$ in line 1) using a multi-lingual parser [6] implemented with Tree-sitter [7].

We organize the candidates into two prioritized groups:

- **Primary Candidates**: condition lines that appear **before** the target line and do not have control dependency over it.

- **Secondary Candidates**: condition lines that appear **after** the target but are *in the same enclosing loop*. These can be misleading because constructs like `break`, `return`, or exceptions may terminate execution early, making them appear influential.

We prioritize sampling from primary candidates and fall back to secondary candidates as needed to maintain a total of five negative instances.

### D.2.3 Sampling Task Instances for Information Flow

For information flow, we follow the same structure as in data dependency task instance generation (Section D.2.1). Given a target variable $v_t$, we sample up to five **positive** and five **negative** instances.

**Positive Sampling.** Positive candidates are selected using GetAllInfoFlowSources($v_t$) used in Algorithm 3 at line 10, which returns all variables that contribute to $v$ through transitive explicit or implicit information flow.

**Negative Sampling.** Negative instances are sampled using the same algorithm described in Algorithm 4, but with the data dependency source set replaced by the information flow source set. Specifically, we substitute line 2 of the algorithm with: $D \leftarrow$ GetAllInfoFlowSources($v_t$).

### D.3 Sampling Statistics

Tables 6 and 7 show the comprehensive breakdown of task instances generated in CORE and CORE Lite. They report the distribution of positive and negative samples across all studied languages. In total, CORE contains 12,553 task instances, while CORE Lite includes 1,584.

Table 6: CORE task instance distribution

| Task Type | | C/C++ | Java | Python | Total |
|---|---|---|---|---|---|
| Data Dependency | Pos. | 510 | 564 | 740 | 1,814 |
| | Neg. | 953 | 989 | 858 | 2,800 |
| | Total | 1,463 | 1,553 | 1,598 | 4,614 |
| Control Dependency | Pos. | 527 | 574 | 592 | 1,693 |
| | Neg. | 628 | 680 | 651 | 1,959 |
| | Total | 1,155 | 1,254 | 1,243 | 3,652 |
| Information Flow | Pos. | 742 | 750 | 799 | 2,291 |
| | Neg. | 641 | 688 | 667 | 1,996 |
| | Total | 1,383 | 1,438 | 1,466 | 4,287 |
| Total | Pos. | 1,779 | 1,888 | 2,131 | 5,798 |
| | Neg. | 2,222 | 2,357 | 2,176 | 6,755 |
| | Total | 4,001 | 4,245 | 4,307 | 12,553 |

Table 7: CORE Lite task instance distribution

| Task Type | | C/C++ | Java | Python | Total |
|---|---|---|---|---|---|
| Data Dependency | Pos. | 69 | 61 | 79 | 209 |
| | Neg. | 131 | 145 | 105 | 381 |
| | Total | 200 | 206 | 184 | 590 |
| Control Dependency | Pos. | 72 | 81 | 86 | 239 |
| | Neg. | 85 | 86 | 79 | 250 |
| | Total | 157 | 167 | 165 | 489 |
| Information Flow | Pos. | 92 | 94 | 105 | 291 |
| | Neg. | 75 | 73 | 65 | 214 |
| | Total | 168 | 167 | 170 | 505 |
| Total | Pos. | 233 | 236 | 270 | 739 |
| | Neg. | 292 | 304 | 249 | 845 |
| | Total | 525 | 540 | 519 | 1,584 |

Table 8: Ablation Study: Semantics-Aware Sampling vs. Random Sampling

| Language | Semantics-Aware Sampling | Random Sampling | | Language | Semantics-Aware Sampling | Random Sampling |
|---|---|---|---|---|---|---|
| C | 82.84 | 83.86 | | C | 31.27 | 37.14 |
| Java | 82.63 | 85.78 | | Java | 31.47 | 37.67 |
| Python | 81.61 | 85.06 | | Python | 29.29 | 31.25 |

F1 for Dependency Classification          Correct Trace Rate for Trace Prediction

## D.4 Ablation Study of the Sampling Strategy

To investigate the impact of the semantics-aware sampling strategy, we conducted an experiment where we removed all heuristics and randomly sampled 400 task instances for the information flow tasks. The performance of Claude 3.5 on this randomly sampled dataset, compared to our original settings, is shown in Table 8.

As observed, performance is consistently better with random sampling. This indicates that random sampling tends to yield easier tasks, as it does not enforce the same level of reasoning complexity as our semantics-aware approach. It is important to note that different sampling strategies result in completely different datasets, and thus, a direct comparison of model performance across these datasets does not necessarily reflect a model's inherent ability.

## E Model Invocation Details

We invoke all models through their respective APIs or platforms using official or authorized access. Below are the details for each model:

- **Claude 3.7 Sonnet** [99], **Claude 3.5 Sonnet** [100], **LLaMA3.1 (405B)** [101], and **DeepSeek R1** [102] are accessed via **AWS Bedrock** [8] through model IDs:
  - Claude 3.7 Sonnet: `us.anthropic.claude-3-7-sonnet-20250219-v1:0`. We enable *thinking mode* with a thinking token budget of 1,024.
  - Claude 3.5 Sonnet: `anthropic.claude-3-5-sonnet-20240620-v1:0`
  - Llama 3.1 405B: `meta.llama3-1-405b-instruct-v1:0`
  - DeepSeek R1: `us.deepseek.r1-v1:0`
- **DeepSeek V3** [103] is accessed through the official **DeepSeek platform** [9] with model ID `deepseek-chat`.
- **Gemini 2.5 Pro** [104] is invoked via the Google's **Gemini Developer API**[10], using model ID `gemini-2.5-pro-preview-05-06`.

---

[6]`https://github.com/PurCL/LLMSCAN`

[7]`https://github.com/tree-sitter/tree-sitter`

[8]`https://aws.amazon.com/bedrock`

[9]`https://www.deepseek.com/`

[10]`https://ai.google.dev/gemini-api/docs`

- **GPT 4o** [105], **GPT o4-mini** [106], and **GPT o3** [106] are accessed through the **OpenAI** [11], with model IDs `gpt-4o`, `o4-mini`, and `o3`.
- **Qwen3 (235B)** [107] is accessed via **OpenRouter** [12] with ID `qwen/qwen3-235b-a22b`.

All models are queried with a decoding temperature of 0 to minimize randomness, except GPT o3 and GPT o4-mini, which do not support zero temperature and are run with the only setting they support: temperature as one. The output length for all models is capped at 2,048 tokens.

Each model is expected to return a response in the predefined JSON format described in the prompts from Section J. If the response is ill-formed or unparsable, we re-query the model using a fallback prompt requesting only the structured output. We retry up to three times, after which the instance is marked as invalid if parsing still fails.

> Fallback Prompt
>
> Your previous response could not be parsed correctly. Please re-read the prompt and ensure your answer strictly follows the required JSON format enclosed with ```<your response here>```.Ensure that your JSON is valid and matches the specification. Try again:

# F  Evaluation Metrics

## F.1  Dependency Classification Metrics

For dependency classification, we report standard metrics: **precision**, **recall**, and **F1 score**, defined as follows:

$$\text{Precision} = \frac{\text{TP}}{\text{TP} + \text{FP}}$$

$$\text{Recall} = \frac{\text{TP}}{\text{TP} + \text{FN}}$$

$$\text{F1 Score} = \frac{2 \cdot \text{Precision} \cdot \text{Recall}}{\text{Precision} + \text{Recall}}$$

We define:

- **True Positive (TP)**: the model predicts that a dependency exists, and the instance is labeled positive.
- **False Positive (FP)**: the model predicts a dependency, but the instance is labeled negative.
- **False Negative (FN)**: the model predicts no dependency, but the instance is labeled positive.

Each task instance has a binary ground truth label indicating whether the specified dependency relation exists between the given pair of program elements.

## F.2  Trace Evaluation Metrics

For task instances where the model predicts a dependency exists, it is also required to return a *trace*: a transitive sequence of direct dependencies (or flows) between program elements. We evaluate the trace at the **edge level**, where each edge is defined as a consecutive ordered pair (`src → dst`) in the predicted trace.

We classify each predicted edge into one of three categories:

- **Valid Edge (VE)**: The edge corresponds to a true direct dependency in the program.
- **Gap Edge**: The source and target nodes are connected by a valid transitive path, but there is no direct edge `src → dst`in the program. This indicates that one or more intermediate nodes were skipped by the model (e.g., `src → x → dst` exists, but x is omitted).

---

[11]`https://platform.openai.com/docs/models`
[12]`https://openrouter.ai/qwen/qwen3-235b-a22b`

- **Invalid Edge (IE)**: There is no valid dependency path, neither direct nor transitive, between `src` and `dst`.

For each predicted trace, we compute the following trace-level metrics:

$$\text{VE}_{\text{trace}} = \frac{\#\text{Valid Edges}}{\#\text{Total Edges in trace}}$$

$$\text{IE}_{\text{trace}} = \frac{\#\text{Invalid Edges}}{\#\text{Total Edges in trace}}$$

$$\text{Missing Steps}_{\text{trace}} = \sum_{e \in \text{gap edges}} \#\text{ intermediate nodes skipped for e}$$

The final reported metrics are the averages over all predicted traces:

$$\text{Valid Edge Rate (VE)} = \frac{1}{N}\sum_{i=1}^{N}\text{VE}_{\text{trace}_i}$$

$$\text{Invalid Edge Rate (IE)} = \frac{1}{N}\sum_{i=1}^{N}\text{IE}_{\text{trace}_i}$$

$$\text{Average Missing Steps (Miss.)} = \frac{1}{N}\sum_{i=1}^{N}\text{Missing Steps}_{\text{trace}_i}$$

$$\text{Correct Trace Rate (CT)} = \frac{\#\text{Correct Traces}}{N}$$

where $N$ is the total number of predicted traces, i.e., the number of task instances where the model predicts the dependency exists.

A perfect trace would consist only of valid direct edges, yielding VE $= 100\%$, IE $= 0\%$, and Miss. $= 0$, and contributes to the numerator of CT.

### F.3 Enumeration Task Metrics

In the enumeration task, the model is asked to list all program elements (variables or lines) that have a specific dependency relation (data dependency, control dependency, or information flow) over a given target element. This is a set prediction task, and we evaluate the model's output using the following metrics:

- **Exact Match (EM):** The prediction is considered correct only if the set of predicted elements exactly matches the ground truth set. Formally,

$$\text{EM} = \begin{cases} 1 & \text{if } \hat{S} = S \\ 0 & \text{otherwise} \end{cases}$$

  where $S$ is the ground-truth set, and $\hat{S}$ is the predicted set.
- **Precision (P):** The fraction of predicted elements that are correct:

$$\text{P} = \frac{|\hat{S} \cap S|}{|\hat{S}|}$$

- **Recall (R):** The fraction of ground-truth elements that are correctly predicted:

$$\text{R} = \frac{|\hat{S} \cap S|}{|S|}$$

- **F1 Score:** The harmonic mean of precision and recall:

$$\text{F1} = \frac{2 \cdot \text{P} \cdot \text{R}}{\text{P} + \text{R}}$$

Table 9: Complete dependency classification results reported as Precision, Recall, and F1 score (%).

| Model | Lang. | CORE Data Dependency P | R | F1 | Control Dependency P | R | F1 | Information Flow P | R | F1 | CORE Lite Data Dependency P | R | F1 | Control Dependency P | R | F1 | Information Flow P | R | F1 |
|---|---|---|---|---|---|---|---|---|---|---|---|---|---|---|---|---|---|---|---|
| Claude 3.7 | C/C++ | 64.76 | 80.00 | 71.58 | 89.12 | 88.61 | 88.87 | 91.74 | 71.83 | 80.57 | 67.90 | 79.71 | 73.33 | 88.00 | 91.67 | 89.80 | 95.77 | 73.91 | 83.44 |
| | Java | 68.11 | 83.69 | 75.10 | 86.72 | 89.90 | 88.28 | 88.71 | 75.47 | 81.56 | 67.44 | 95.08 | 78.91 | 90.12 | 90.12 | 90.12 | 90.12 | 77.66 | 83.43 |
| | Python | 76.49 | 79.59 | 78.01 | 84.82 | 86.82 | 85.81 | 91.61 | 68.34 | 78.28 | 74.16 | 83.54 | 78.57 | 91.03 | 82.56 | 86.59 | 90.00 | 68.57 | 77.84 |
| | Average | 69.79 | 81.09 | 74.90 | 86.89 | 88.44 | 87.65 | 90.69 | 71.88 | 80.14 | 69.83 | 86.11 | 76.94 | 89.72 | 88.12 | 88.84 | 91.96 | 73.38 | 81.57 |
| Claude 3.5 | C/C++ | 53.90 | 90.78 | 67.64 | 69.98 | 88.05 | 77.98 | 82.40 | 83.29 | 82.84 | 56.64 | 92.75 | 70.33 | 70.33 | 88.89 | 78.53 | 84.78 | 84.78 | 84.78 |
| | Java | 54.80 | 90.07 | 68.14 | 65.62 | 91.46 | 76.42 | 80.69 | 84.67 | 82.63 | 52.21 | 96.72 | 67.82 | 69.81 | 91.36 | 79.14 | 85.71 | 82.98 | 84.32 |
| | Python | 64.99 | 88.78 | 75.04 | 67.06 | 86.99 | 75.74 | 86.21 | 77.47 | 81.61 | 62.07 | 91.14 | 73.85 | 72.00 | 83.72 | 77.42 | 90.00 | 77.14 | 83.08 |
| | Average | 57.90 | 89.88 | 70.27 | 67.55 | 88.83 | 76.71 | 83.10 | 81.81 | 82.36 | 56.97 | 93.54 | 70.67 | 70.71 | 87.99 | 78.36 | 86.83 | 81.63 | 84.06 |
| DeepSeek R1 | C/C++ | 84.92 | 80.59 | 82.70 | 94.94 | 88.99 | 91.87 | 96.71 | 75.20 | 84.61 | 87.30 | 79.71 | 83.33 | 95.71 | 93.06 | 94.37 | 96.20 | 82.61 | 88.89 |
| | Java | 84.07 | 81.38 | 82.70 | 96.67 | 90.94 | 93.72 | 92.37 | 72.67 | 81.34 | 84.13 | 86.89 | 85.48 | 95.00 | 93.83 | 94.41 | 94.37 | 71.28 | 81.21 |
| | Python | 86.43 | 76.62 | 81.23 | 95.14 | 85.98 | 90.33 | 94.15 | 72.47 | 81.90 | 83.78 | 78.48 | 81.05 | 95.89 | 81.40 | 88.05 | 96.05 | 69.52 | 80.66 |
| | Average | 85.14 | 79.53 | 82.21 | 95.58 | 88.64 | 91.97 | 94.41 | 73.45 | 82.62 | 85.07 | 81.69 | 83.29 | 95.53 | 89.43 | 92.28 | 95.54 | 74.47 | 83.59 |
| DeepSeek V3 | C/C++ | 60.79 | 81.76 | 69.73 | 74.78 | 79.89 | 77.25 | 84.59 | 72.51 | 78.08 | 65.43 | 76.81 | 70.67 | 72.62 | 84.72 | 78.21 | 85.37 | 76.09 | 80.46 |
| | Java | 63.81 | 84.40 | 72.67 | 73.61 | 83.10 | 78.07 | 82.83 | 69.47 | 75.56 | 60.87 | 91.80 | 73.20 | 77.27 | 83.95 | 80.47 | 85.92 | 64.89 | 73.94 |
| | Python | 74.06 | 82.57 | 78.08 | 70.74 | 77.20 | 73.83 | 85.42 | 68.21 | 75.85 | 67.42 | 75.95 | 71.43 | 81.71 | 77.91 | 79.76 | 90.79 | 65.71 | 76.24 |
| | Average | 66.22 | 82.91 | 73.49 | 73.04 | 80.06 | 76.38 | 84.28 | 70.06 | 76.50 | 64.57 | 81.52 | 71.77 | 77.20 | 82.19 | 79.48 | 87.36 | 68.90 | 76.88 |
| Gemini 2.5 Pro | C/C++ | 81.36 | 94.12 | 87.27 | 86.36 | 97.34 | 91.53 | 95.89 | 94.34 | 95.11 | 81.01 | 92.75 | 86.49 | 86.42 | 97.22 | 91.50 | 90.91 | 97.83 | 94.24 |
| | Java | 86.07 | 92.02 | 88.95 | 91.64 | 97.39 | 94.43 | 94.50 | 96.13 | 95.31 | 83.10 | 96.72 | 89.39 | 89.89 | 98.77 | 94.12 | 97.80 | 94.68 | 96.22 |
| | Python | 87.90 | 93.24 | 90.49 | 89.22 | 95.10 | 92.07 | 94.55 | 95.62 | 95.08 | 86.05 | 93.67 | 89.70 | 91.86 | 91.86 | 91.86 | 92.59 | 95.24 | 93.90 |
| | Average | 85.11 | 93.13 | 88.90 | 89.07 | 96.61 | 92.68 | 94.98 | 95.36 | 95.17 | 83.39 | 94.38 | 88.53 | 89.39 | 95.95 | 92.49 | 93.77 | 95.92 | 94.79 |
| GPT o3 | C/C++ | 92.87 | 91.96 | 92.41 | 94.06 | 90.13 | 92.05 | 97.81 | 90.43 | 93.98 | 94.03 | 91.30 | 92.65 | 91.89 | 94.44 | 93.15 | 95.45 | 91.30 | 93.33 |
| | Java | 93.51 | 94.50 | 94.00 | 96.35 | 91.99 | 94.12 | 94.53 | 92.13 | 93.32 | 95.16 | 96.72 | 95.93 | 93.75 | 92.59 | 93.17 | 94.38 | 89.36 | 91.80 |
| | Python | 94.18 | 91.89 | 93.02 | 96.65 | 87.67 | 91.94 | 95.24 | 90.24 | 92.67 | 91.14 | 91.14 | 91.14 | 97.30 | 83.72 | 90.00 | 93.37 | 89.52 | 91.26 |
| | Average | 93.52 | 92.78 | 93.14 | 95.69 | 89.93 | 92.70 | 95.86 | 90.93 | 93.32 | 93.44 | 93.05 | 93.24 | 94.31 | 90.25 | 92.11 | 94.30 | 90.06 | 92.13 |
| GPT o4-mini | C/C++ | 89.86 | 76.47 | 82.63 | 94.09 | 87.67 | 90.77 | 95.85 | 74.80 | 84.03 | 94.64 | 76.81 | 84.80 | 93.06 | 93.06 | 93.06 | 96.10 | 80.43 | 87.57 |
| | Java | 87.92 | 74.82 | 80.84 | 97.90 | 89.20 | 93.35 | 93.98 | 74.93 | 83.38 | 92.31 | 78.69 | 84.96 | 97.37 | 91.36 | 94.27 | 94.94 | 79.79 | 86.71 |
| | Python | 92.26 | 74.05 | 82.16 | 96.34 | 84.46 | 90.01 | 96.17 | 72.34 | 82.57 | 91.18 | 78.48 | 84.35 | 97.22 | 81.40 | 88.61 | 95.83 | 65.71 | 77.97 |
| | Average | 90.01 | 75.11 | 81.88 | 96.11 | 87.11 | 91.38 | 95.33 | 74.02 | 83.33 | 92.71 | 77.99 | 84.70 | 95.88 | 88.61 | 91.98 | 95.62 | 75.31 | 84.08 |
| GPT 4o | C/C++ | 53.21 | 89.41 | 66.72 | 72.08 | 77.42 | 74.66 | 71.02 | 87.87 | 78.55 | 55.65 | 92.75 | 69.57 | 69.77 | 83.33 | 75.95 | 70.34 | 90.22 | 79.05 |
| | Java | 57.03 | 87.77 | 69.13 | 71.08 | 81.36 | 75.87 | 66.95 | 83.47 | 74.30 | 51.89 | 90.16 | 65.87 | 70.97 | 81.48 | 75.86 | 69.30 | 84.04 | 75.96 |
| | Python | 67.56 | 85.00 | 75.28 | 74.07 | 73.82 | 73.94 | 72.05 | 80.35 | 75.98 | 63.89 | 87.34 | 73.80 | 82.50 | 76.74 | 79.52 | 77.55 | 72.38 | 74.88 |
| | Average | 59.27 | 87.39 | 70.38 | 72.41 | 77.53 | 74.82 | 70.01 | 83.90 | 76.28 | 57.14 | 90.08 | 69.75 | 74.41 | 80.52 | 77.11 | 72.40 | 82.21 | 76.63 |
| Llama 3.1 405B | C/C++ | 46.09 | 79.80 | 58.44 | 63.16 | 79.70 | 70.47 | 73.05 | 74.53 | 73.78 | 48.39 | 86.96 | 62.18 | 66.28 | 79.17 | 72.15 | 74.47 | 76.09 | 75.27 |
| | Java | 49.05 | 82.45 | 61.51 | 59.21 | 80.66 | 68.29 | 68.54 | 77.87 | 72.91 | 45.24 | 93.44 | 60.96 | 64.42 | 82.72 | 72.43 | 69.39 | 72.34 | 70.83 |
| | Python | 56.98 | 78.92 | 66.18 | 59.04 | 76.69 | 66.72 | 70.33 | 73.59 | 71.93 | 56.76 | 79.75 | 66.32 | 61.39 | 72.09 | 66.31 | 76.64 | 78.10 | 77.36 |
| | Average | 50.71 | 80.39 | 62.04 | 60.47 | 79.02 | 68.49 | 70.64 | 75.33 | 72.87 | 50.13 | 86.72 | 63.15 | 64.03 | 77.99 | 70.30 | 73.50 | 75.51 | 74.49 |
| Qwen3 235B | C/C++ | 83.76 | 76.86 | 80.16 | 93.51 | 81.97 | 87.36 | 97.79 | 53.64 | 69.28 | 84.06 | 84.06 | 84.06 | 96.77 | 83.33 | 89.55 | 96.43 | 58.70 | 72.97 |
| | Java | 82.66 | 72.70 | 77.36 | 93.70 | 85.54 | 89.44 | 92.99 | 54.80 | 68.96 | 84.21 | 78.69 | 81.36 | 96.05 | 90.12 | 92.99 | 96.15 | 53.19 | 68.49 |
| | Python | 86.10 | 73.65 | 79.39 | 91.90 | 76.69 | 83.61 | 95.97 | 56.70 | 71.28 | 84.93 | 78.48 | 81.58 | 90.67 | 79.07 | 84.47 | 96.43 | 51.43 | 67.08 |
| | Average | 84.17 | 74.40 | 78.97 | 93.04 | 81.40 | 86.80 | 95.58 | 55.05 | 69.84 | 84.40 | 80.41 | 82.33 | 94.50 | 84.17 | 89.00 | 96.34 | 54.44 | 69.51 |

These metrics are computed per instance and then averaged across all enumeration queries in the benchmark. Exact match provides a strict correctness measure, while precision, recall, and F1 allow partial credit and provide more granular insight into model behavior.

# G   Complete Evaluation Results

Complete evaluation results for CORE and CORE Lite are provided in Table 9 (dependency classification), Table 10 (trace generation), and Table 11 (dependency source enumeration).

**CORE vs. CORE Lite**   We observe minimal performance differences between CORE and CORE Lite, suggesting that CORE Lite serves as a reliable alternative when computational resources are limited.

**Performance Analysis on Task Types.**   For trace generation 10, we observe that information flow tasks consistently exhibit significantly higher average missing steps (Miss.) and invalid edge (IE) rates compared to data and control dependency task types. This suggests that information flow is the most challenging task for LLMs, where they frequently generate incorrect edges. A comparison between Claude 3.5 and Claude 3.7 reveals that while their valid edge rates (VE) are similar, Claude 3.7 demonstrates a much lower invalid edge rate (IE). This indicates Claude 3.7 possesses better reasoning capabilities, leading to fewer erroneous edges.

Regarding dependency source enumeration (Table 11), different models display distinct error tendencies. For models such as Claude 3.7, DeepSeek R1, Llama 3.1 (405B), Qwen 3 (235B), GPT 4-mini, and GPT 4o, recall is consistently lower than precision across all task types (data dependency, control dependency, and information flow). This pattern suggests these models tend to yield precise dependency sources but frequently miss many relevant ones. Conversely, models like Gemini 2.5 Pro and DeepSeek V3 do not exhibit such a consistent trend in their precision-recall balance.

Table 10: Complete trace evaluation results: Correct Trace Rate (CT, %), Valid Edge Rate (VE, %), Invalid Edge Rate (IE, %), and Avg. Missing Steps (Miss). ↑ means higher is better; ↓ lower is better.

| Model | Lang. | CORE Data Dependency CT↑ | VE↑ | IE↓ | Miss.↓ | Control Dependency CT↑ | VE↑ | IE↓ | Miss.↓ | Information Flow CT↑ | VE↑ | IE↓ | Miss.↓ | CORE Lite Data Dependency CT↑ | VE↑ | IE↓ | Miss.↓ | Control Dependency CT↑ | VE↑ | IE↓ | Miss.↓ | Information Flow CT↑ | VE↑ | IE↓ | Miss.↓ |
|---|---|---|---|---|---|---|---|---|---|---|---|---|---|---|---|---|---|---|---|---|---|---|---|---|---|
| Claude 3.7 | C/C++ | 66.47 | 71.33 | 6.82 | 0.05 | 58.06 | 65.43 | 2.86 | 0.39 | 38.81 | 53.72 | 8.34 | 0.32 | 72.46 | 74.46 | 3.67 | 0.04 | 62.50 | 69.26 | 2.27 | 0.35 | 41.30 | 57.78 | 6.67 | 0.27 |
| | Java | 60.64 | 69.43 | 9.85 | 0.11 | 59.06 | 68.36 | 3.84 | 0.39 | 35.20 | 55.59 | 10.51 | 0.40 | 72.13 | 77.62 | 7.90 | 0.16 | 62.96 | 73.27 | 2.04 | 0.33 | 39.36 | 59.16 | 10.31 | 0.35 |
| | Python | 57.03 | 67.17 | 7.35 | 0.20 | 48.99 | 58.66 | 3.09 | 0.59 | 37.80 | 53.57 | 7.18 | 0.34 | 63.29 | 73.29 | 8.16 | 0.15 | 54.65 | 62.46 | 3.34 | 0.36 | 39.05 | 54.38 | 6.86 | 0.29 |
| | Average | 61.38 | 69.31 | 8.01 | 0.12 | 55.37 | 64.15 | 3.26 | 0.46 | 37.27 | 54.29 | 8.68 | 0.35 | 69.29 | 75.12 | 6.58 | 0.12 | 60.04 | 68.33 | 2.55 | 0.35 | 39.90 | 57.11 | 7.95 | 0.30 |
| Claude 3.5 | C/C++ | 50.39 | 65.67 | 23.38 | 0.06 | 55.79 | 67.98 | 9.51 | 0.24 | 31.27 | 55.29 | 15.67 | 0.52 | 72.75 | 69.79 | 20.91 | 0.06 | 56.94 | 70.80 | 6.20 | 0.28 | 40.22 | 61.47 | 11.42 | 0.57 |
| | Java | 44.86 | 62.10 | 24.14 | 0.12 | 53.83 | 70.18 | 11.99 | 0.27 | 31.47 | 58.03 | 16.00 | 0.60 | 60.66 | 78.93 | 15.27 | 0.10 | 59.26 | 72.47 | 9.36 | 0.28 | 35.11 | 59.24 | 14.03 | 0.79 |
| | Python | 42.30 | 60.11 | 24.62 | 0.20 | 48.99 | 65.11 | 9.86 | 0.40 | 29.29 | 52.20 | 14.84 | 0.53 | 46.84 | 64.77 | 22.03 | 0.25 | 55.81 | 72.23 | 5.21 | 0.28 | 32.38 | 55.76 | 11.42 | 0.37 |
| | Average | 45.85 | 62.63 | 24.05 | 0.13 | 52.87 | 67.76 | 10.45 | 0.30 | 30.68 | 55.17 | 15.50 | 0.55 | 54.19 | 71.16 | 19.40 | 0.14 | 57.34 | 71.83 | 6.92 | 0.28 | 35.90 | 58.82 | 12.29 | 0.57 |
| DeepSeek R1 | C/C++ | 71.96 | 74.78 | 5.11 | 0.01 | 67.17 | 76.17 | 1.06 | 0.31 | 41.64 | 55.69 | 6.68 | 0.40 | 69.57 | 71.50 | 7.49 | 0.01 | 73.61 | 83.75 | 1.39 | 0.28 | 40.22 | 58.48 | 4.17 | 0.53 |
| | Java | 68.09 | 73.70 | 3.92 | 0.10 | 68.82 | 78.39 | 1.06 | 0.35 | 39.07 | 55.29 | 5.46 | 0.45 | 75.41 | 78.80 | 3.99 | 0.13 | 71.60 | 81.28 | 0.62 | 0.37 | 40.43 | 55.63 | 5.20 | 0.39 |
| | Python | 57.57 | 65.92 | 3.84 | 0.27 | 57.94 | 69.55 | 0.82 | 0.51 | 35.79 | 53.79 | 5.05 | 0.54 | 56.96 | 67.85 | 4.14 | 0.34 | 54.65 | 66.24 | 0.39 | 0.42 | 38.10 | 52.76 | 5.43 | 0.45 |
| | Average | 65.87 | 71.47 | 4.29 | 0.13 | 64.64 | 74.70 | 0.98 | 0.39 | 38.83 | 54.92 | 5.73 | 0.46 | 67.31 | 72.72 | 5.21 | 0.16 | 66.62 | 77.09 | 0.80 | 0.36 | 39.58 | 55.62 | 4.93 | 0.46 |
| DeepSeek V3 | C/C++ | 56.27 | 63.99 | 13.91 | 0.07 | 43.83 | 55.80 | 7.77 | 0.38 | 22.64 | 41.10 | 21.65 | 0.39 | 60.87 | 66.11 | 5.99 | 0.07 | 48.61 | 63.43 | 7.27 | 0.35 | 21.74 | 41.58 | 24.09 | 0.37 |
| | Java | 52.13 | 63.00 | 14.73 | 0.19 | 37.80 | 52.72 | 9.05 | 0.53 | 22.40 | 42.56 | 16.46 | 0.52 | 63.93 | 73.14 | 13.96 | 0.10 | 41.98 | 56.58 | 4.32 | 0.57 | 24.47 | 41.44 | 14.68 | 0.53 |
| | Python | 45.41 | 57.03 | 14.57 | 0.43 | 32.09 | 43.88 | 7.52 | 0.63 | 19.65 | 37.10 | 19.29 | 0.60 | 44.30 | 54.99 | 12.71 | 0.35 | 30.23 | 40.81 | 7.40 | 0.78 | 24.76 | 40.41 | 14.91 | 0.50 |
| | Average | 51.27 | 61.34 | 14.40 | 0.23 | 37.91 | 50.80 | 8.11 | 0.51 | 21.56 | 40.25 | 19.13 | 0.51 | 56.37 | 64.75 | 10.89 | 0.18 | 40.27 | 53.61 | 6.33 | 0.56 | 23.66 | 41.14 | 17.89 | 0.47 |
| Gemini 2.5 Pro | C/C++ | 92.16 | 92.92 | 0.90 | 0.00 | 93.74 | 95.79 | 0.22 | 0.03 | 69.14 | 84.33 | 1.31 | 0.31 | 92.75 | 92.75 | 0.00 | 0.00 | 95.83 | 96.94 | 0.00 | 0.01 | 68.48 | 86.11 | 0.63 | 0.39 |
| | Java | 83.16 | 87.82 | 2.81 | 0.06 | 92.51 | 95.28 | 0.06 | 0.07 | 68.40 | 85.26 | 1.38 | 0.37 | 88.52 | 93.85 | 2.32 | 0.03 | 92.59 | 95.23 | 0.00 | 0.11 | 67.02 | 83.40 | 0.35 | 0.39 |
| | Python | 86.08 | 91.14 | 1.14 | 0.05 | 89.53 | 92.70 | 0.51 | 0.06 | 67.83 | 85.87 | 1.46 | 0.35 | 89.87 | 92.83 | 0.32 | 0.03 | 88.37 | 90.31 | 0.39 | 0.02 | 70.48 | 86.38 | 1.59 | 0.32 |
| | Average | 87.13 | 90.63 | 1.62 | 0.04 | 91.93 | 94.59 | 0.26 | 0.05 | 68.46 | 85.15 | 1.38 | 0.34 | 90.38 | 93.14 | 0.88 | 0.02 | 92.26 | 94.16 | 0.13 | 0.05 | 68.66 | 85.30 | 0.86 | 0.37 |
| GPT o3 | C/C++ | 89.61 | 90.47 | 0.85 | 0.01 | 79.70 | 84.63 | 0.60 | 0.12 | 52.56 | 72.21 | 3.70 | 0.42 | 88.41 | 88.89 | 0.97 | 0.01 | 86.11 | 90.00 | 0.00 | 0.12 | 56.52 | 76.56 | 0.72 | 0.41 |
| | Java | 83.16 | 88.98 | 1.89 | 0.11 | 80.84 | 86.23 | 0.00 | 0.17 | 51.07 | 72.91 | 4.42 | 0.57 | 91.80 | 94.13 | 0.00 | 0.07 | 79.01 | 85.91 | 0.00 | 0.19 | 51.06 | 71.01 | 5.59 | 0.51 |
| | Python | 78.92 | 86.81 | 0.79 | 0.16 | 71.28 | 78.54 | 0.11 | 0.26 | 49.19 | 72.08 | 3.43 | 0.57 | 78.48 | 87.04 | 1.27 | 0.16 | 67.44 | 74.77 | 0.00 | 0.21 | 49.52 | 72.17 | 3.42 | 0.52 |
| | Average | 83.90 | 88.75 | 1.18 | 0.10 | 77.27 | 83.13 | 0.24 | 0.19 | 50.94 | 72.40 | 3.85 | 0.52 | 86.23 | 90.02 | 0.75 | 0.08 | 77.52 | 83.56 | 0.00 | 0.17 | 52.37 | 73.25 | 3.24 | 0.48 |
| GPT o4-mini | C/C++ | 73.73 | 74.40 | 0.75 | 0.02 | 67.55 | 75.44 | 0.23 | 0.29 | 40.97 | 55.47 | 1.70 | 0.41 | 75.36 | 75.36 | 1.45 | 0.00 | 69.44 | 79.17 | 0.00 | 0.32 | 42.39 | 60.29 | 2.36 | 0.43 |
| | Java | 62.77 | 67.35 | 2.26 | 0.13 | 67.94 | 76.18 | 0.41 | 0.37 | 41.07 | 55.79 | 2.71 | 0.48 | 73.77 | 75.41 | 0.00 | 0.10 | 74.07 | 81.54 | 0.41 | 0.31 | 43.62 | 58.92 | 2.77 | 0.62 |
| | Python | 59.59 | 66.71 | 1.40 | 0.20 | 56.08 | 66.31 | 0.00 | 0.49 | 41.93 | 57.17 | 1.57 | 0.44 | 62.03 | 72.16 | 2.15 | 0.18 | 56.98 | 66.05 | 0.00 | 0.43 | 40.95 | 51.57 | 2.38 | 0.36 |
| | Average | 65.36 | 69.49 | 1.47 | 0.12 | 63.86 | 72.64 | 0.21 | 0.38 | 41.32 | 56.14 | 1.99 | 0.44 | 70.39 | 74.31 | 1.20 | 0.09 | 66.83 | 75.59 | 0.14 | 0.35 | 42.32 | 56.93 | 2.50 | 0.47 |
| GPT 4o | C/C++ | 60.39 | 66.18 | 15.66 | 0.12 | 41.56 | 51.95 | 8.49 | 0.38 | 23.72 | 44.56 | 29.31 | 0.66 | 60.87 | 65.16 | 17.45 | 0.14 | 43.06 | 56.81 | 5.76 | 0.51 | 26.09 | 49.96 | 24.29 | 0.70 |
| | Java | 55.67 | 64.73 | 13.39 | 0.23 | 40.94 | 53.33 | 7.62 | 0.53 | 23.47 | 46.36 | 24.09 | 0.64 | 63.93 | 67.08 | 12.57 | 0.20 | 45.68 | 54.65 | 5.43 | 0.53 | 23.40 | 49.58 | 21.74 | 0.70 |
| | Python | 52.30 | 61.90 | 13.01 | 0.33 | 35.47 | 47.07 | 5.51 | 0.55 | 20.15 | 41.81 | 24.43 | 0.82 | 60.76 | 70.53 | 6.22 | 0.42 | 40.70 | 54.26 | 5.23 | 0.56 | 18.10 | 38.18 | 22.26 | 0.83 |
| | Average | 56.12 | 64.27 | 14.02 | 0.23 | 39.32 | 50.78 | 7.21 | 0.49 | 22.45 | 44.24 | 25.94 | 0.71 | 61.85 | 67.59 | 12.08 | 0.25 | 43.15 | 55.24 | 5.47 | 0.53 | 22.53 | 45.91 | 22.76 | 0.74 |
| Llama 3.1 405B | C/C++ | 37.45 | 45.42 | 28.39 | 0.10 | 31.50 | 42.60 | 19.03 | 0.39 | 13.75 | 27.63 | 36.93 | 0.41 | 34.78 | 43.69 | 37.84 | 0.14 | 30.56 | 41.47 | 20.11 | 0.31 | 13.04 | 24.65 | 39.47 | 0.45 |
| | Java | 32.80 | 44.21 | 30.06 | 0.21 | 26.13 | 39.13 | 17.96 | 0.56 | 15.73 | 33.16 | 31.98 | 0.58 | 50.82 | 61.17 | 25.30 | 0.18 | 27.16 | 38.17 | 13.95 | 0.60 | 14.89 | 33.31 | 27.63 | 0.64 |
| | Python | 27.30 | 37.82 | 31.66 | 0.31 | 25.17 | 37.15 | 17.81 | 0.62 | 15.02 | 28.65 | 32.48 | 0.67 | 32.91 | 41.43 | 29.60 | 0.41 | 25.58 | 33.99 | 13.33 | 0.64 | 20.00 | 35.11 | 28.82 | 0.77 |
| | Average | 32.52 | 42.48 | 30.04 | 0.21 | 27.60 | 39.63 | 18.27 | 0.52 | 14.83 | 29.81 | 33.80 | 0.55 | 39.50 | 48.76 | 30.91 | 0.24 | 27.77 | 37.88 | 15.80 | 0.52 | 15.98 | 31.02 | 31.97 | 0.62 |
| Qwen3 235B | C/C++ | 68.43 | 70.74 | 4.85 | 0.02 | 59.39 | 68.04 | 2.12 | 0.30 | 29.78 | 39.82 | 4.78 | 0.27 | 73.91 | 76.81 | 7.25 | 0.00 | 61.11 | 73.10 | 1.62 | 0.28 | 31.52 | 44.58 | 2.97 | 0.32 |
| | Java | 60.11 | 64.80 | 4.60 | 0.09 | 58.01 | 68.05 | 3.23 | 0.38 | 28.13 | 39.81 | 5.71 | 0.35 | 65.57 | 70.36 | 4.23 | 0.11 | 62.96 | 78.17 | 1.26 | 0.40 | 26.60 | 39.40 | 5.04 | 0.38 |
| | Python | 55.27 | 63.43 | 4.03 | 0.24 | 48.14 | 58.02 | 1.62 | 0.49 | 28.66 | 41.59 | 5.96 | 0.35 | 56.96 | 65.62 | 6.75 | 0.33 | 53.49 | 60.66 | 0.68 | 0.48 | 31.43 | 38.91 | 6.65 | 0.23 |
| | Average | 61.27 | 66.32 | 4.49 | 0.11 | 55.18 | 64.70 | 2.32 | 0.39 | 28.86 | 40.41 | 5.48 | 0.32 | 65.48 | 70.93 | 6.08 | 0.15 | 59.19 | 68.96 | 1.52 | 0.38 | 29.85 | 40.96 | 4.89 | 0.31 |

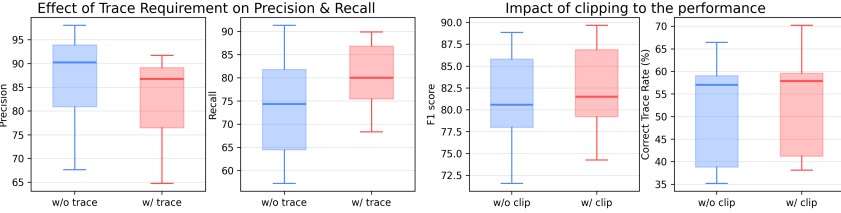

(a) F1 scores for dependency classification    (b) Correct trace rates for trace generation

Figure 4: Impact of query type (classification only vs. with trace) and input clipping on F1 score and correct trace rate across tasks.

**Performance Analysis on Programming Languages.** Overall, we do not observe a significant performance trend across programming languages for dependency classification (Table 9) or trace generation (Table 10). This holds across task types (data dependency, control dependency, and information flow) and models, with no evidence that any language is consistently more challenging or that any model is particularly strong in one language. This is likely because performance is heavily influenced by the specific structure of each program.

However, for the dependency source enumeration task (Table 11), we observe a consistent pattern in most of the cases: models perform notably better on C and Java than on Python, with performance gaps sometimes substantial. We attribute this to the higher information density of Python due to the lack of explicit block delimiters (e.g., {}) and the use of syntactic sugar. The absence of typing annotations makes precise reasoning more difficult.

# H    The Impact of Different Experimental Settings on Model Behaviors

We conduct a few-shot learning (FSL) study using both sparse (BM25) and dense (CodeBERT [108]) retrievers to retrieve the top-$k$ most similar examples from the rest of the entire dataset, with $k \in \{4, 8, 12, 16, 20\}$. Retrieved examples are sorted by similarity score, with the most relevant placed closest to the query in the prompt. We evaluate this setup using Claude 3.7 in thinking mode, focusing on the pairwise dependency classification task. For each query, the model is asked whether the dependency holds, and if so, to generate a valid trace. To mitigate label imbalance and reduce bias, we ensure an equal number of positive and negative examples in each prompt.

Table 11: Complete dependency source enumeration results reported as Exact Match Rate (EM), Precision (P), Recall (R), and F1 score (%).

| Model | Lang. | CORE Lite | | | | | | | | | | | |
|---|---|---|---|---|---|---|---|---|---|---|---|---|---|
| | | Data Dependency | | | | Control Dependency | | | | Information Flow | | | |
| | | EM | P | R | F1 | EM | P | R | F1 | EM | P | R | F1 |
| Claude 3.7 | C/C++ | 23.00 | 75.35 | 64.10 | 59.94 | 50.96 | 98.04 | 81.32 | 85.40 | 10.12 | 90.27 | 57.25 | 64.11 |
| | Java | 16.99 | 75.07 | 60.90 | 57.31 | 59.28 | 97.62 | 87.01 | 90.21 | 3.59 | 91.93 | 59.56 | 67.39 |
| | Python | 21.74 | 81.93 | 65.59 | 67.55 | 44.85 | 96.29 | 78.74 | 83.81 | 7.06 | 89.12 | 59.29 | 66.88 |
| | Average | 20.58 | 77.45 | 63.53 | 61.60 | 51.70 | 97.32 | 82.36 | 86.47 | 6.92 | 90.44 | 58.70 | 66.13 |
| Claude 3.5 | C/C++ | 13.50 | 55.01 | 73.95 | 57.89 | 40.13 | 89.65 | 79.49 | 80.81 | 10.71 | 84.37 | 61.15 | 67.12 |
| | Java | 10.68 | 62.88 | 75.41 | 62.14 | 47.31 | 91.00 | 86.13 | 86.25 | 4.79 | 89.16 | 59.14 | 67.74 |
| | Python | 9.24 | 64.06 | 60.47 | 56.39 | 33.94 | 90.08 | 80.21 | 81.68 | 7.65 | 87.40 | 58.70 | 66.15 |
| | Average | 11.14 | 60.65 | 69.94 | 58.81 | 40.46 | 90.24 | 81.94 | 82.91 | 7.72 | 86.98 | 59.66 | 67.00 |
| DeepSeek R1 | C/C++ | 46.50 | 90.24 | 75.35 | 77.84 | 56.05 | 98.68 | 80.59 | 85.84 | 8.93 | 92.03 | 48.93 | 59.63 |
| | Java | 45.15 | 92.32 | 72.38 | 76.20 | 52.10 | 98.30 | 79.89 | 86.10 | 4.79 | 90.23 | 44.47 | 55.26 |
| | Python | 25.00 | 89.35 | 59.01 | 65.67 | 36.97 | 97.80 | 72.57 | 80.27 | 7.65 | 93.41 | 41.64 | 52.70 |
| | Average | 38.88 | 90.64 | 68.91 | 73.24 | 48.37 | 98.26 | 77.68 | 84.07 | 7.12 | 91.89 | 45.01 | 55.86 |
| DeepSeek V3 | C/C++ | 20.00 | 64.24 | 66.86 | 58.53 | 26.11 | 79.61 | 68.37 | 69.31 | 3.57 | 66.60 | 62.59 | 60.09 |
| | Java | 16.99 | 68.09 | 56.93 | 54.21 | 24.55 | 81.73 | 75.79 | 74.86 | 2.40 | 72.52 | 58.55 | 60.19 |
| | Python | 10.87 | 71.06 | 56.89 | 57.29 | 13.33 | 76.35 | 62.97 | 65.21 | 1.76 | 75.08 | 56.90 | 59.69 |
| | Average | 15.95 | 67.80 | 60.23 | 56.68 | 21.33 | 79.23 | 69.04 | 69.79 | 2.58 | 71.40 | 59.35 | 59.99 |
| Gemini 2.5 Pro | C/C++ | 53.50 | 82.87 | 96.31 | 85.46 | 76.43 | 95.09 | 94.18 | 92.61 | 35.71 | 93.90 | 87.88 | 88.07 |
| | Java | 43.69 | 82.89 | 95.41 | 84.80 | 79.64 | 96.08 | 96.22 | 95.12 | 20.36 | 92.84 | 88.55 | 88.45 |
| | Python | 51.09 | 87.12 | 94.14 | 87.41 | 70.91 | 92.45 | 95.65 | 92.53 | 24.12 | 93.70 | 92.70 | 92.00 |
| | Average | 49.43 | 84.29 | 95.29 | 85.89 | 75.66 | 94.54 | 95.35 | 93.42 | 26.73 | 93.48 | 89.71 | 89.51 |
| GPT o3 | C/C++ | 53.00 | 89.54 | 91.63 | 88.55 | 68.79 | 98.30 | 87.41 | 90.16 | 18.45 | 95.14 | 78.84 | 84.39 |
| | Java | 42.23 | 87.57 | 91.49 | 86.95 | 77.25 | 98.88 | 90.93 | 93.28 | 8.38 | 91.52 | 77.24 | 80.32 |
| | Python | 30.43 | 88.64 | 90.08 | 87.05 | 66.67 | 98.34 | 87.80 | 90.94 | 20.00 | 96.62 | 79.50 | 84.43 |
| | Average | 41.89 | 88.58 | 91.07 | 87.52 | 70.90 | 98.51 | 88.71 | 91.46 | 15.61 | 94.43 | 78.53 | 83.05 |
| GPT o4-mini | C/C++ | 34.50 | 86.97 | 81.04 | 79.11 | 63.06 | 98.95 | 85.01 | 89.20 | 11.31 | 95.15 | 56.05 | 66.03 |
| | Java | 35.44 | 89.01 | 71.52 | 71.65 | 66.47 | 99.64 | 86.46 | 90.67 | 5.99 | 95.05 | 50.31 | 61.39 |
| | Python | 17.39 | 82.74 | 66.79 | 68.59 | 55.76 | 98.96 | 80.64 | 86.08 | 9.41 | 95.15 | 49.52 | 60.35 |
| | Average | 29.11 | 86.24 | 73.12 | 73.12 | 61.76 | 99.18 | 84.04 | 88.65 | 8.90 | 95.12 | 51.96 | 62.59 |
| GPT 4o | C/C++ | 14.00 | 64.52 | 51.70 | 48.65 | 31.85 | 86.53 | 70.07 | 74.10 | 4.17 | 73.60 | 46.18 | 53.19 |
| | Java | 13.59 | 71.22 | 52.91 | 51.59 | 34.13 | 85.21 | 74.42 | 76.43 | 1.80 | 80.76 | 37.35 | 46.61 |
| | Python | 9.24 | 74.97 | 48.58 | 53.95 | 21.21 | 85.17 | 70.74 | 73.23 | 2.94 | 80.85 | 35.18 | 44.90 |
| | Average | 12.28 | 70.24 | 51.06 | 51.40 | 29.06 | 85.64 | 71.74 | 74.59 | 2.97 | 78.40 | 39.57 | 48.23 |
| Llama 3.1 405B | C/C++ | 2.50 | 60.28 | 29.48 | 23.84 | 7.64 | 61.33 | 45.11 | 43.59 | 3.57 | 66.61 | 28.78 | 33.91 |
| | Java | 5.83 | 59.15 | 29.52 | 26.21 | 5.99 | 64.25 | 42.85 | 42.25 | 1.20 | 67.67 | 25.31 | 30.83 |
| | Python | 0.54 | 56.56 | 15.25 | 15.33 | 2.42 | 57.58 | 36.12 | 36.88 | 0.00 | 59.55 | 23.13 | 28.41 |
| | Average | 2.96 | 58.66 | 24.75 | 21.79 | 5.35 | 61.05 | 41.36 | 40.91 | 1.59 | 64.61 | 25.74 | 31.05 |
| Qwen3 235B | C/C++ | 25.50 | 95.93 | 34.94 | 34.83 | 47.77 | 96.84 | 70.24 | 73.82 | 5.95 | 97.87 | 21.23 | 25.28 |
| | Java | 22.82 | 94.40 | 35.94 | 36.93 | 51.50 | 96.59 | 75.02 | 78.82 | 1.80 | 96.38 | 20.36 | 25.60 |
| | Python | 15.76 | 95.58 | 28.14 | 28.97 | 34.55 | 95.13 | 63.34 | 66.90 | 6.47 | 96.96 | 18.55 | 22.88 |
| | Average | 21.36 | 95.30 | 33.01 | 33.58 | 44.61 | 96.19 | 69.53 | 73.18 | 4.74 | 97.07 | 20.05 | 24.59 |

As shown in Fig. 5, performance drops as the number of retrieved few-shot examples ($K$) increases for both dependency classification and trace generation. This degradation supports our hypothesis that longer inputs dilute model attention from task instructions and that retrieved examples, despite being label-balanced, can introduce bias. Furthermore, standard retrieval methods (e.g., BM25, CodeBERT) struggle to capture task-relevant structural and semantic similarity, highlighting the need for more retrieval strategies tailored to code reasoning.

# I  Other experimental explorations

## I.1  Prompt with Reverse Dependency Examples

In Section 5.2, the experimental results shows that LLMs struggle with queries that involve reverse dependency order. To further explore the capabilities of LLMs in dealing with such complex dependencies, we added an example of reverse dependency reasoning to the prompt and conducted experiments with Claude 3.5 on data dependency tasks. The results are shown in Table 12.

The results show negligible differences with no significant improvement or decrease. However, our primary contribution is providing a robust dataset rather than optimizing model performance through prompt engineering. Including examples or things to pay attention to for all challenging scenarios

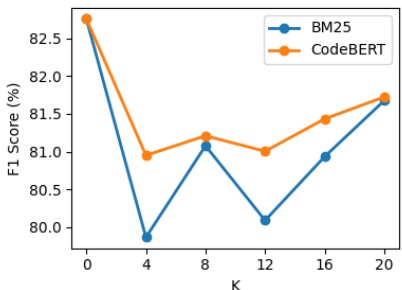
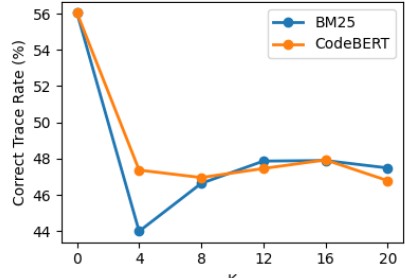

(a) F1 scores for dependency classification    (b) Correct trace rates for trace generation

Figure 5: F1 and correct trace rate under FSL as the number of retrieved examples ($k$) increases.

Table 12: Effect of adding reverse dependency reasoning examples to the prompts on dependency classification and trace generation tasks.

| Language | Original | Modified Prompt |
|----------|----------|-----------------|
| C | 67.64 | 66.86 |
| Java | 68.14 | 68.34 |
| Python | 75.04 | 74.60 |

F1 for Dependency Classification

| Language | Original | Modified Prompt |
|----------|----------|-----------------|
| C | 50.39 | 50.00 |
| Java | 44.86 | 45.57 |
| Python | 42.30 | 39.32 |

Correct Trace Rate for Trace Generation

would be impractical and could be considered a form of overfitting to specific cases rather than evaluating genuine reasoning capabilities.

## I.2    Shorten the Prompt

Our prompts (section J) are designed to be concise while including necessary definitions and illustrative examples, which are particularly important for non-reasoning models and align with common few-shot learning practices [61]. Modern LLMs also support very long input contexts, making prompt length less of a constraint.

To investigate the impact of prompt length, we conducted an experiment where we shortened the prompts by cutting half of the examples. The results for Claude 3.5 on data dependency tasks are in Table 13. These results show negligible overall differences, with a slight decrease in some metrics.

## J    Prompt Design

In this section, we include the prompt templates used for both pairwise dependency queries and target-centric dependency source enumeration (Section 3.3.1, covering data dependency, control dependency, and information flow. For clarity, content dynamically generated per data point, such as function bodies or variable names, is shown in blue, while the rest represents the fixed prompt structure.

---

**Prompt for Pairwise Dependency Query (Data Dependency)**

You are a program analysis assistant. Perform a **static** data dependence analysis on a given code snippet, treating each branch or loop condition as potentially taking any outcome, without using semantic or symbolic execution to prune paths.

**## 1. Data Dependence Definition**
Data dependence captures the influence of data flow between variables.
We denote each variable instance as `(var,lineNumber)`, meaning the variable `var` defined or updated at `lineNumber`.
**Direct Data Dependence**
A variable instance `(varB,lineB)` is **directly** data dependent on `(varA,lineA)` if `lineA` writes to `varA`. Then, without overwritting the value of `varA`, `lineB` reads `varA` and computes the value of `varB`. In other words, the value of `(varB,lineB)` relies on the value produced at `(varA,lineA)`. In addition, changing the execution order of `(varA,lineA)` and `(varB,lineB)` would alter their results.
A variable instance `(varB,lineB)` is data dependent on `(varA,lineA)` if there is a **transitive** (indirect) chain of **direct** data dependencies from `(varA,lineA)` to `(varB,lineB)`. This is equivalent to saying that `(varA,lineA)` has data dependence over

---

Table 13: Effect of prompt length reduction on dependency classification and trace generation tasks.

| Language | Original | Shortened |
|---|---|---|
| C | 67.64 | 67.50 |
| Java | 68.14 | 68.17 |
| Python | 75.04 | 74.04 |

F1 for Dependency Classification

| Language | Original | Shortened |
|---|---|---|
| C | 50.39 | 49.80 |
| Java | 44.86 | 46.63 |
| Python | 42.30 | 39.86 |

Correct Trace Rate for Trace Generation

---

(`varB,lineB`), meaning the value written at (`varA,lineA`) can propagate to (`varB,lineB`) through a sequence of read/write operations.

## 2. Output Format
When asked "Does (`varA,lineA`) have data dependence over (`varB,lineB`)? If so, provide one feasible trace.":

- If data dependence exists:
```json
{
 "DataDependence": true,
 "Trace": [
{ "from": ["varA", lineA], "to": ["varX", lineX] },
...,
{ "from": ["varY", lineY], "to": ["varB", lineB] }
]
}
```

- If no data dependence exists:
```json
{
 "DataDependence": false
}
```

A **trace** represents a **transitive data flow**, where each item is a **direct** data dependence edge. The full trace should begin with (`varA,lineA`) and end with (`varB,lineB`). Only **one** valid chain is required even if multiple chains are possible.

## 3. Intraprocedural Data Dependence
All dependence analysis is performed within **a single function**. We do not track dependencies across function boundaries. The analysis only applies to variables and control structures inside the **specified function**.

## 4. Example Code Snippet
### Example 1
```python
1    total = 0
2    value = 1
3    step = value
4    if step > 1:
5        value += 3
6    while total <= 10:
7        total += value
8        difference = total - step
9        step += 1
10   final_result = total * 2
```

#### Example Question 1.1:
Does (`value,2`) have data dependence over (`step,9`)? If so, provide a trace.
**Analysis**:
– Line 9: step += 1 ⇒ (`step,9`) directly depends on (`step,3`) (or itself).
– Line 3: step = value ⇒ (`step,3`) directly depends on (`value,2`).
Hence the chain (`value,2`) → (`step,3`) → (`step,9`).
 Output:
```json
{
 "DataDependence": true,
 "Trace": [
{ "from": ["value", 2], "to": ["step", 3] },
{ "from": ["step", 3], "to": ["step", 9] }
]
}
```

#### Example Question 1.2:
Does (`value,2`) have data dependence over (`total,7`)? If so, provide a trace.
**Analysis**:
– Line 7: total += value reads value from (`value,2`) or (`value,5`).
– Line 5: value += 3 depends on (`value,2`) but may be conditionally executed.
Providing one feasible chain: (`value,2`) → (`value,5`) → (`total,7`).

Output:
```json
{
 "DataDependence": true,
 "Trace": [
{ "from": ["value", 2], "to": ["value", 5] },
{ "from": ["value", 5], "to": ["total", 7] }
 ]
}
```

#### Example Question 1.3:
 Does (`step`,3) have data dependence over (`final_result`,10)? If so, provide a trace.
 **Analysis**: No direct or transitive dependence exists.
  Output:
```json
{
 "DataDependence": false
}
```

### Example 2
```python
1    arr = [1, 2, 3]
2    x = 5
3    if x > 2:
4        arr[0] = x
5    for i in range(3):
6        arr[i] += 1
7        temp = arr[i]
8        y = temp + 1
9    result = arr[-1]
```
#### Example Question 2.1:
 Does (`x`,2) have data dependence over (`arr`,6)? If so, provide a trace.
 **Analysis**: One feasible chain is (`x`,2) $\rightarrow$ (`arr`,4) $\rightarrow$ (`arr`,6).
  Output:
```json
{
 "DataDependence": true,
 "Trace": [
{ "from": ["x", 2], "to": ["arr", 4] },
{ "from": ["arr", 4], "to": ["arr", 6] }
 ]
}
```

#### Example Question 2.2:
 Does (`i`,5) have data dependence over (`temp`,7)? If so, provide a trace.
 **Analysis**: Direct dependence (`i`,5) $\rightarrow$ (`temp`,7).
  Output:
```json
{
 "DataDependence": true,
 "Trace": [
{ "from": ["i", 5], "to": ["temp", 7] }
 ]
}
```

#### Example Question 2.3:
 Does (`i`,5) have data dependence over (`result`,9)? If so, provide a trace.
 **Analysis**: Feasible chain (`i`,5) $\rightarrow$ (`arr`,6) $\rightarrow$ (`result`,9).
  Output:
```json
{
 "DataDependence": true,
 "Trace": [
{ "from": ["i", 5], "to": ["arr", 6] },
{ "from": ["arr", 6], "to": ["result", 9] }
 ]
}
```

 [YOUR TURN]
 Below is **your target snippet**.
```
<target code with line number>
```
 **Question**:    Does    (`src_var, src_line`)    have    data    dependence    over    (`dst_var, dst_line`)    in    function
`target_function_name`? If so, provide a trace.
 **Output**:

You are a program analysis assistant. Perform a **static** data dependence analysis on a given code snippet, treating each branch or loop condition as potentially taking any outcome, without using semantic or symbolic execution to prune paths.

## 1. Data Dependence Definition

Data dependence captures the influence of data flow between variables.

We denote each variable instance as (`var`,`lineNumber`), meaning the variable `var` defined or updated at `lineNumber`.

**Direct Data Dependence**

A variable instance (`varB`,`lineB`) is **directly** data dependent on (`varA`, `lineA`) if `lineA` writes to `varA`. Then, without overwriting the value of `varA`, `lineB` reads `varA` and computes the value of `varB`. In other words, the value of (`varB`,`lineB`) relies on the value produced at (`varA`,`lineA`). In addition, changing the execution order of (`varA`,`lineA`) and (`varB`,`lineB`) would alter their results.

A variable instance (`varB`,`lineB`) is data dependent on (`varA`,`lineA`) if there is a **transitive** (indirect) chain of **direct** data dependencies from (`varA`,`lineA`) to (`varB`,`lineB`). This is equivalent to saying that (`varA`,`lineA`) has data dependence over (`varB`,`lineB`), meaning the value written at (`varA`,`lineA`) can propagate to (`varB`,`lineB`) through a sequence of read/write operations.

## 2. Output Format

When asked "Which variable instances have data dependence over (`targetVar`, `targetLine`)? List all such variables.", respond:

```json
{
"DataDependenceSources": [
["valA", lineA],
["valB", lineB],
...
]
}
```

If you believe there are no variables with data dependence over (`targetVar`, `targetLine`), respond:

```json
{
"DataDependenceSources": [ ]
}
```

## 3. Intraprocedural Data Dependence

All dependence analysis is performed within **a single function**. We do not track dependencies across function boundaries. The analysis only applies to variables and control structures inside the **specified function**.

## 4. Example Code Snippet

### Example 1

```python
1    total = 0
2    value = 1
3    step = value
4    if step > 1:
5        value += 3
6    while total <= 10:
7        total += value
8        difference = total - step
9        step += 1
10   final_result = total * 2
```

#### Example Question 1.1:

Which variable instances have data dependence over (`step`,9)? List all such variables.

**Analysis**:

– Line 9: `step += 1` ⇒ (`step`,9) directly depends on (`step`,3) (or on itself).

– Line 3: `step = value` ⇒ (`step`,3) directly depends on (`value`,2).

Hence the transitive chain is (`value`,2) → (`step`,3) → (`step`,9).

Output:

```json
{
"DataDependenceSources": [
["step", 9],
["step", 3],
["value", 2],
]
}
```

#### Example Question 1.2:

Which variable instances have data dependence over (`total`,7)? List all such variables.

**Analysis**:

– Line 7: `total += value` reads `total` (from line 1 or itself) and `value` (from line 2 or 5).

– Line 5: `value += 3` depends on (`value`,2) but is under an `if` guard; static analysis assumes both execution paths.

Therefore possible sources are (`total`,1), (`total`,7), (`value`,5), and (`value`,2).

Output:

```json
{
"DataDependenceSources": [
["total", 1],
```

```
["total", 7],
["value", 5],
["value", 2]
]
}
```

### Example 2
```python
1    arr = [1, 2, 3]
2    x = 5
3    if x > 2:
4        arr[0] = x
5    for i in range(3):
6        arr[i] += 1
7        temp = arr[i]
8        y = temp + 1
9    result = arr[-1]
```
#### Example Question 2.1:
Which variable instances have data dependence over (arr,6)? List all such variables.
**Analysis**:
– Line 6 modifies arr[i]. If i == 0, it relies on (arr,4); otherwise on (arr,1).
– Line 4 depends on (x,2).
– i itself originates at line 5.
  Output:
```json
{
"DataDependenceSources": [
["arr", 6],
["arr", 4],
["arr", 1],
["i", 5],
["x", 2]
]
}
```

#### Example Question 2.2:
Which variable instances have data dependence over (temp,7)? List all such variables.
**Analysis**:
– Line 7: temp = arr[i] ⇒ depends on arr[i] (from line 6) and on i (line 5).
  Output:
```json
{
"DataDependenceSources": [
["arr", 6],
["i", 5],
]
}
```

#### Example Question 2.3:
Which variable instances have data dependence over (result,9)? List all such variables.
**Analysis**:
– Line 9: result = arr[-1] where arr[-1] (arr[2]) may come from initialization (line 1) or updates (line 6), which in turn
depend on i (line 5).
  Output:
```json
{
"DataDependenceSources": [
["arr", 6],
["arr", 1],
["i", 5],
]
}
```
[YOUR TURN]
Below is **your target snippet**.
```

<target code with line number>
```

**Question**: Which variable instances have data dependence over (dst_var, dst_line) in function target_function_name?
List all such variables.
**Output**:

You are a program-analysis assistant. Your task is to statically analyze the control dependence of a given code snippet.

## 1. Control Dependence Definition

Control dependence captures the influence of control flow decisions on the execution of statements.

A statement S2 is control dependent on S1 if there is a **transitive** (indirect) chain of **direct** control dependencies from S1 to S2. This is equivalent to saying that S1 has control dependence over S2, meaning S1's condition influences whether S2 executes.

**Direct Control Dependence:**

A statement S2 is **directly** control dependent on a statement S1 if

1. S1 is a conditional control statement (e.g. an `if`, `while`, `for`, `switch`, etc.).

2. S1 directly determines whether S2 executes. That is, S1 has multiple successor branches,

   - there exists **at least one branch** in which S2 **always executes**, and

   - there exists **at least one other branch** in which S2 does **not necessarily execute**.

"Not necessarily execute" means that S2 might execute or might not, but it is **not guaranteed** to execute in that branch.

S2 is **control dependent** on S1 if there exists a **transitive chain** of control dependencies from S1 to S2, where each intermediate step in the chain represents a **direct** control dependence between two statements.

## 2. Output Format

When asked, "Does line S1 have control dependence over line S2? If so, provide a trace." you should respond in JSON format as follows:

- If there *is* a control dependence:
  ```json
  {
  "ControlDependence":  true,
  "Trace":  [S1, ..., S2]
  }
  ```

- If there is *no* control dependence:
  ```json
  {
  "ControlDependence":  false
  }
  ```

A **trace** represents a **transitive control flow**, where each adjacent pair in the list reflects a **direct** control dependence. The trace must start with S1 and end with S2.

## 3. Interprocedural Control Dependence

All dependence analysis is performed within **a single function**. We do not track dependencies across function boundaries. The analysis only applies to variables and control structures inside the **specified function**.

## 4. Example Code Snippet

### Example 1

```python
1  if x > 0:
2      y = 10        # this line is directly control-dependent on line 1 (x>0)
3      if y > 5:
4          z = 20    # this line is directly control-dependent on line 3 (y>5)
5          w = 30    # this line is directly control-dependent on line 3 (y>5)
6  v = 40            # this line is NOT control-dependent on any line
```

**Analysis**:

– Line 2 is directly control dependent on line 1 (x>0).

– Lines 4 and 5 are directly control dependent on line 3 (y>5).

– Lines 4 and 5 are **indirectly** control dependent on line 1 (x>0) because even if line 1 (x>0) is `true`, lines 4 and 5 may not execute if line 3's condition (y>5) is `false`. However, since control dependence is transitive, lines 4 and 5 are **indirectly** control dependent on line 1.

– Line 6 is not control dependent on any lines because it always executes, regardless of whether line 1 (x>0) or line 3 (y>5) is true or false.

#### Example Question 1.1:

Does line 1 have control dependence over line 5? If so, provide a trace.

 Output:
```json
{
"ControlDependence": true,
"Trace": [1, 3, 5]
}
```

#### Example Question 1.2:

Does line 3 have control dependence over line 6? If so, provide a trace.

 Output:
```json
{
"ControlDependence": false
}
```

### Example 2

```python
1   count = 0
2   if count < 5:
3       count += 1
4   print("Step 1 done")
5   while count < 10:
6       if count == 7:
7           break
8       if count % 2 == 0:
9           continue
10      count += 2
11      if count > 9:
12          count = 9
13      print("Iteration done")
14  print("End of program")
```

#### Example Question 2.1:
Does line 5 have control dependence over line 13? If so, provide a trace.
**Analysis**:
– Line 13 is directly control dependent on line 8 because if line 8 evaluates to `true`, the `continue` skips line 13 for that iteration.
– Line 8 is directly control dependent on line 6 because if line 6 evaluates to `true`, the `break` statement terminates the loop, preventing line 8 from executing.
– Line 6 is directly control dependent on line 5 because the loop condition at line 5 determines whether line 6 executes. If line 5 evaluates to `false`, execution skips the loop entirely.

Since control dependence is transitive, line 5 indirectly controls line 13 through lines 6 and 8. The trace is 5→6→8→13.
  Output:
```json
{
"ControlDependence": true,
"Trace": [5, 6, 8, 13]
}
```

#### Example Question 2.2:
Does line 8 have control dependence over line 10? If so, provide a trace.
**Analysis**: Similar to the explanation above, line 8 is an `if` condition inside the `while` loop with a `continue`, so it can skip the rest of the loop entirely, including line 10. Therefore, line 10 is directly control dependent on 8.
  Output:
```json
{
"ControlDependence": true,
"Trace": [8, 10]
}
```

#### Example Question 2.3:
Does line 5 have control dependence over line 14? If so, provide a trace.
**Analysis**: Line 5 does not affect whether line 14 is reached. Line 14 is outside the `while` loop (lines 5–13) and will be executed regardless of the condition.
  Output:
```json
{
"ControlDependence": false
}
```

#### Example Question 2.4:
Does line 2 have control dependence over line 12? If so, provide a trace.
**Analysis**: Line 2 does not affect whether line 12 is reached. The condition in line 2 only controls whether line 3 is executed. Line 12 is inside the `while` loop (lines 5–13) and may or may not execute regardless of the condition in line 2.
  Output:
```json
{
"ControlDependence": false
}
```
 [YOUR TURN]
Below is **your target snippet**.
```
<target code with line number>
```
**Question**: Does `src_line` have control dependence over `dst_line` in function `target_function_name`? If so, provide a trace.
**Output**:

Prompt for Dependency Source Enumeration (Control Dependency)

You are a program-analysis assistant. Your task is to statically analyze the control dependence of a given code snippet.
## 1. Control Dependence Definition
Control dependence captures the influence of control-flow decisions on the execution of statements.

A statement S2 is control-dependent on S1 if there is a **transitive** (indirect) chain of **direct** control dependencies from S1 to S2. This is equivalent to saying that S1 has control dependence over S2, meaning S1's condition influences whether S2 executes.
**Direct Control Dependence:** A statement S2 is **directly** control-dependent on a statement S1 if:
1. S1 is a conditional control statement (e.g., an `if`, `while`, `for`, `switch`, etc.).
2. S1 directly determines whether S2 executes. That is, S1 has multiple successor branches,
- there exists **at least one branch** in which S2 **always executes**, and - there exists **at least one other branch** in which S2 does **not necessarily execute**.
"Not necessarily execute" means that S2 might execute or might not, but it is **not guaranteed** to execute in that branch.
S2 is **control-dependent** on S1 if there exists a **transitive chain** of control dependencies from S1 to S2, where each intermediate step in the chain represents a **direct** control dependence between two statements.

## 2. Output Format

When asked, "Which lines have control dependence over `targetLine`? List all such lines." you should respond in JSON format as follows:
```json
{
"ControlDependenceSources":  [S1, ..., S2]
}
```
If you believe there are no lines that have control dependence over `targetLine`, respond:
```json
{
"ControlDependenceSources":  []
}
```

## 3. Interprocedural Control Dependence

All dependence analysis is performed within **a single function**. We do not track dependencies across function boundaries. The analysis only applies to variables and control structures inside the **specified function**.

## 4. Example Code Snippet

### Example 1
```python
1 if x > 0:
2     y = 10 # this line is directly control-dependent on line 1 (x>0)
3     if y > 5:
4         z = 20 # this line is directly control-dependent on line 3 (y>5)
5         w = 30 # this line is directly control-dependent on line 3 (y>5)
6 v = 40 # this line is NOT control-dependent on any line
```

**Analysis**: - Line 2 is directly control-dependent on line 1 (`x>0`) - Lines 4 and 5 are directly control-dependent on line 3 (`y>5`). - Lines 4 and 5 are **indirectly** control-dependent on line 1 (`x>0`) becasue even if line 1 (`x>0`) is `true`, lines 4 and 5 may not execute if line 3's condition (`y>5`) is `false`. However, since control dependence is transitive, lines 4 and 5 are **indirectly** control-dependent on line 1. - Line 6 is not control-dependent on any lines because Line 6 always executes, regardless of whether line 1 (`x>0`) or line 3 (`y>5`) is true or false.

#### Example Question 1.1: Which lines have control dependence over line 5? List all such lines.
Output:
```json
{
"ControlDependenceSources":  [1, 3]
}
```

#### Example Question 1.2: Which lines have control dependence over line 6? List all such lines.
Output:
```json
{
"ControlDependenceSources":  []
}
```

### Example 2
```python
1 count = 0
2 if count < 5:
3     count += 1
4 print("Step 1 done")
5 while count < 10:
6     if count == 7:
7         break
8     if count % 2 == 0:
9         continue
10    count += 2
11    if count > 9:
12        count = 9
13    print("Iteration done")
14 print("End of program")
```

#### Example Question 2.1: Which lines have control dependence over line 13? List all such lines.
**Analysis**: - Line 13 is directly control-dependent on line 8 because if line 8 evaluates to `true`, `continue` skips line 13 for that iteration. - Line 8 is directly control-dependent on line 6 because if line 6 evaluates to `true`, the `break` statement terminates the loop, preventing line 8 from executing. - Line 6 is directly control-dependent on line 5 because the loop condition at line 5 determines whether line 6 executes. If line 5 evaluates to `false`, execution skips the loop entirely.
Output:
```json
```

```
{
 "ControlDependenceSources":  [5, 6, 8]
}
```
#### Example Question 2.2: Which lines have control dependence over line 10? List all such lines.
**Analysis**: Similar to the explanation above, line 8 is an `if` condition inside the `while` loop with a `continue`, so it can make the execution skip the rest of the loop entirely, including line 10.
 Output:
```json
{
 "ControlDependenceSources":  [5, 6, 8]
}
```
#### Example Question 2.3: Which lines have control dependence over line 14? List all such lines.
**Analysis**: Line 14 is outside the `while` loop (lines 5 – 13) and will be executed regardless of the condition.
 Output:
```json
{
 "ControlDependenceSources":  []
}
```
#### Example Question 2.4: Which lines have control dependence over line 12? List all such lines.
**Analysis**: Line 2 doesn't affect whether line 12 is reached. The condition in line 2 only controls whether line 3 is executed. Line 12 is in the `while` loop (lines 5 – 13) and will or will not be executed regardless of the condition in line 2.
 Output:
```json
{
 "ControlDependenceSources":  [5, 6, 8, 11]
}
```
[YOUR TURN]
Below is **your target snippet**.
```
<target code with line number>
```
**Question**: Which lines have control dependence over `dst_line` in function `target_function_name`? List all such lines.
**Output**:

Prompt for Pairwise Dependency Query (Information Flow)

You are a program-analysis assistant. Please perform a **static** analysis of "information flow" on a given code snippet, treating each branch or loop condition as potentially taking any outcome, without using semantic or symbolic execution to prune paths.

## 1. Information Flow Definition
An **information flow** from variable x to variable y, written x -> y, exists whenever information stored in x is transferred to or used to derive information transferred to y.
We denote each variable instance as (var,lineNumber), meaning the variable var is **defined or updated** at lineNumber.

### Types of Direct Flows

- **Direct Explicit Flow**
  A direct explicit flow occurs when the value written by one variable instance is directly used in computing the value of another, through operations like assignments.

  Formally, if (varA, lineA) writes a value and (varB, lineB) reads that value to compute its own, then there is a **direct explicit flow** from (varA, lineA) to (varB, lineB), written as: (varA,lineA) -> (varB,lineB).

  **Example 1**:
  ```python
  1   x = 5
  2   y = x + 1
  ```
  (x,1) -> (y,2) is a direct explicit flow.

  **Example 2**:
  ```python
  1   x = 5
  2   if cond:
  3       y = x + 1
  ```
  (x,1) -> (y,3) is a direct explicit flow.

  Under static analysis, we treat **each conditional or loop as potentially taking any outcome**, and we **do not perform semantic pruning**. Therefore, even if the statement at line 3 is conditionally executed, we still recognize (x,1) as used to compute (y,3).

- **Direct Implicit Flow**
  A direct implicit flow from (varA,lineA) to (varB, lineB) occurs when (varA,lineA)'s value directly influences whether lineB executes. In other words, it corresponds to a **direct control dependence**, where the condition has **at least one branch** where (varB, lineB) must run and another where it may not. All conditional structures (e.g. if, while, for, switch, etc.) generate these flows.

  Formally, if (varA,lineA) is **directly read** in a condition at lineC and that condition **directly** decides whether lineB executes, then we say there is a **direct implicit flow** from (varA,lineA) to (varB,lineB) (written (varA,lineA) -> (varB,lineB)).

**Example**:
```python
1  z = x + 5
2  if z > 1:
3      y = 10
```

`(z,1) -> (z,2) -> (y,3)` is a direct implicit flow through the condition at line 2. The asterisk indicates that the variable is used for the conditional statement but not redefined at that line (i.e. used in a condition).

### Information Flow Between Variables
An **information flow** exists from a variable (`varA`, `lineA`) to another variable (`varB`, `lineB`) if there is a **transitive chain of direct flows** — explicit, implicit, or both — connecting them:
`(varA, lineA) -> ... -> (varB, lineB)`

## 2. Output Format
When asked "Is there information flow from (`varA`,`lineA`) to (`varB`,`lineB`)? If so, provide one feasible trace.", respond with a JSON object:

- If information flow exists:
```json
{
"InformationFlow": true,
"Trace": [
{
"from": ["varOrExpr", lineNumber        /*, "use" optional */],
"to":   ["varOrExpr", lineNumber        /*, "use" optional */],
"type": "data" | "control"
},
...
]
}
```

- If no information flow exists:
```json
{
"InformationFlow": false
}
```

Every trace edge must specify `"from"`, `"to"` (each may include a third `"use"` element when the line is a conditional statement that only reads without redefining/updating the value), and `"type"`. **We do not consider loop in the trace**.
A **trace** represents a **transitive chain of direct flows** arising from compositions of **direct** implicit or explicit flows, where each edge represents a **direct implicit flow** (through control dependence) or a **direct explicit flow** (through data dependence).
You only need to provide **one** valid trace if you conclude an information flow exists, even if multiple possible chains exist.

## 3. Intraprocedural Data Dependence
All dependence analysis is performed within **a single function**. We do not track dependencies across function boundaries. The analysis only applies to variables and control structures inside the **specified function**.

## 4. Example Code Snippet
### Example 1
```python
1  def example_func():
2      status = 0
3      flag = False
4      balance = 1000
5      balance += 500
6      if balance > 1000:
7          status = 1
8          flag = True
9      limit = status * 5000
10     transaction = limit * 0.2
11     return flag
```
#### Example Question 1.1:
Is there information flow from (`balance`,4) to (`transaction`,10)? If so, provide a trace.
**Analysis**:
– Line 10: `transaction` reads `limit`; direct explicit flow (`limit`,9) -> (`transaction`,10).
– Line 9: `limit` reads `status`; direct explicit flow (`status`,7) -> (`limit`,9).
– Line 7: assignment executed only if line 6 condition true; control-dependent on line 6.
– Line 6: condition `balance > 1000` reads `balance`; implicit flow (`balance`,5) -> (`status`,7).
– Line 5: `balance` updated from (`balance`,4); explicit flow (`balance`,4) -> (`balance`,5).
  Output:
```json
{
"InformationFlow": true,
"Trace": [
{ "from": ["balance", 4], "to": ["balance", 5], "type": "data" },
{ "from": ["balance", 5], "to": ["balance", 6, "use"], "type": "data" },
{ "from": ["balance", 6, "use"], "to": ["status", 7], "type": "control" },
{ "from": ["status", 7], "to": ["limit", 9], "type": "data" },
```

```json
{ "from": ["limit", 9], "to": ["transaction", 10], "type": "data" }
]
}
```

#### Example Question 1.2:
Is there information flow from (`flag`,3) to (`limit`,9)? If so, provide a trace.
**Analysis**: No transitive chain (explicit or implicit) connects (`flag`,3) to (`limit`,9).
 Output:
```json
{
"InformationFlow": false
}
```

#### Example Question 1.3:
Is there information flow from (`status`,2) to (`transaction`,10)? If so, provide a trace.
**Analysis**: One feasible explicit-flow chain is (`status`,2) -> (`limit`,9) -> (`transaction`,10).
 Output:
```json
{
"InformationFlow": true,
"Trace": [
{ "from": ["status", 2], "to": ["limit", 9], "type": "data" },
{ "from": ["limit", 9], "to": ["transaction", 10], "type": "data" }
]
}
```

### Example 2
```python
1.  val = 5
2.  size = 3
3.  arr = [0] * size
4.  i = 0
5.  j = 1
6.  total = 0
7.  if val > 2:
8.    arr[j % size] = j
9.  while i < size:
10.   arr[j] += 1
11.   score = arr[j+1]
12.   total += score * 2
13.   diff = total - score
14.   j = (j + 1) % size
15.   i += 1
16. last = arr[-1]
17. summary = diff + j
```
#### Example Question 2.1:
Is there information flow from (`size`,2) to (`summary`,17)? If so, provide a trace.
**Analysis**: Feasible chain (`size`,2) -> (`size`,9,"use") -> (`j`,14) -> (`summary`,17).
 Output:
```json
{
"InformationFlow": true,
"Trace": [
{ "from": ["size", 2], "to": ["size", 9, "use"], "type": "data" },
{ "from": ["size", 9, "use"], "to": ["j", 14], "type": "control" },
{ "from": ["j", 14], "to": ["summary", 17], "type": "data" }
]
}
```

#### Example Question 2.2:
Is there information flow from (`j`,5) to (`score`,11)? If so, provide a trace.
**Analysis**: Direct explicit flow (`j`,5) -> (`score`,11) suffices.
 Output:
```json
{
"InformationFlow": true,
"Trace": [
{ "from": ["j", 5], "to": ["score", 11], "type": "data" }
]
}
```

#### Example Question 2.3:
Is there information flow from (`j`,1) to (`i`,15)? If so, provide a trace.
**Analysis**: No explicit or implicit chain links (`j`,1) to (`i`,15).

```
 Output:
```json
{
"InformationFlow": false
}
```
```

#### Example Question 2.4:
Is there information flow from `(val,1)` to `(last,16)`? If so, provide a trace.
**Analysis**: One feasible chain `(val,1) -> (val,7,"use") -> (arr,8) -> (last,16)` exists.
```
 Output:
```json
{
"InformationFlow": true,
"Trace": [
{ "from": ["val", 1], "to": ["val", 7, "use"], "type": "data" },
{ "from": ["val", 7, "use"], "to": ["arr", 8], "type": "control" },
{ "from": ["arr", 8], "to": ["last", 16], "type": "data" }
]
}
```
```

[YOUR TURN]
Below is **your target snippet**.
```
<target code with line number>
```

**Question**:   Is  there  information  flow  from  `(src_var, src_line)`  to  `(dst_var, dst_line)`  in  function
`target_function_name`? If so, provide a trace.
**Output**:

You are a program analysis assistant. Please perform a **static** analysis of "information flow" on a given code snippet, treating each branch or loop condition as potentially taking any outcome, without using semantic or symbolic execution to prune paths.

## 1. Information Flow Definition
An **information flow** from variable `x` to variable `y`, written `x → y`, exists whenever information stored in `x` is transferred to or used to derive information transferred to `y`.
We denote each variable instance as `(var,lineNumber)`, meaning the variable `var` is **defined or updated** at `lineNumber`.

### Types of Direct Flows
- **Direct Explicit Flow** A direct explicit flow occurs when the value written by one variable instance is directly used in computing the value of another, through operations like assignments. Formally, if `(varA, lineA)` writes a value and `(varB, lineB)` reads that value to compute its own, then there is a **direct explicit flow** from `(varA, lineA)` to `(varB, lineB)`, written as: `(varA,lineA) → (varB,lineB)`.

  **Example 1**:
  ```
  1   x = 5
  2   y = x + 1
  ```
  `(x,1) → (y,2)` is a direct explicit flow.

  **Example 2**:
  ```python
  1   x = 5
  2   if cond:
  3     y = x + 1
  ```
  `(x,1) → (y,3)` is still a direct explicit flow since static analysis assumes line 3 may execute.

- **Direct Implicit Flow**
  A direct implicit flow from `(varA,lineA)` to `(varB,lineB)` occurs when `(varA,lineA)`'s value directly influences whether `lineB` executes—that is, there is a **direct control dependence**.

  Formally, if `(varA,lineA)` is read in a condition at `lineC` and that condition directly decides whether `lineB` executes, we write the direct implicit flow as `(varA,lineA) → (varB,lineB)`.

  **Example**:
  ```python
  1   z = x + 5
  2   if z > 1:
  3     y = 10
  ```
  `(z,1) → (z,2)* → (y,3)` is a direct implicit flow through the condition at line 2. The asterisk "*" marks that `z` is *used* at the condition and not re-defined there.

## 2. Output Format
When asked "Which variable instances have information flow over `(targetVar,targetLine)`? List all such variables.", respond:
- With variables present:
  ```json
  {
  "InfomationFlowSources": [
  ["valA", lineA],
```

```
                ["valB", lineB],
                ...
                ]
                }
                """
```

- With none present:
```json
{
"InfomationFlowSources": false
}
```

A flow exists from (`varA`,`lineA`) to (`varB`,`lineB`) if there is any **transitive** chain of direct flows—explicit, implicit, or both—linking them.

## 3. Intraprocedural Data Dependence
All analysis is confined to **a single function**; flows are not tracked across function boundaries.

## 4. Example Code Snippet
### Example 1
```python
1  def example_func():
2    status = 0
3    flag = False
4    balance = 1000
5    balance += 500
6    if balance > 1000:
7      status = 1
8      flag = True
9    limit = status * 5000
10   transaction = limit * 0.2
11   return flag
```

#### Example Question 1.1:
Which variable instances have information flow over (`transaction`,10)? List all such variables.
**Analysis**:
There is an explicit chain: (`limit`,9) $\rightarrow$ (`transaction`,10), then (`status`,7)/(`status`,2) $\rightarrow$ (`limit`,9). Line 7 is control-dependent on the condition at line 6, which reads `balance`. Thus flows propagate from (`balance`,4) and (`balance`,5).
Output:
```json
{
"InfomationFlowSources": [
["limit", 9],
["status", 2],
["status", 7],
["balance", 4],
["balance", 5]
]
}
```

#### Example Question 1.2:
Which variable instances have information flow over (`flag`,3)? List all such variables.
**Analysis**: Line 3 simply initializes `flag`.
Output:
```json
{
"InfomationFlowSources": []
}
```

### Example 2
```python
1.  val = 5
2.  size = 3
3.  arr = [0] * size
4.  i = 0
5.  j = 1
6.  total = 0
7.  if val > 2:
8.    arr[j % size] = j
9.  while i < size:
10.   arr[j] += 1
11.   score = arr[j+1]
12.   total += score * 2
13.   diff = total - score
14.   j = (j + 1) % size
15.   i += 1
16.  last = arr[-1]
17.  summary = diff + j
```

#### Example Question 2.1:
Which variable instances have information flow over (summary,17)? List all such variables.
  Output:
```json
{
"InfomationFlowSources": [
["diff", 13],
["j", 14],
["j", 5],
["size", 2],
["i", 4],
["i", 15]
]
}
```

#### Example Question 2.2:
Which variable instances have information flow over (score,11)? List all such variables.
  Output:
```json
{
"InfomationFlowSources": [
["arr", 10],
["arr", 8],
["arr", 3],
["j", 14],
["j", 5],
["size", 2],
["i", 4],
["i", 15],
["val", 1]
]
}
```

#### Example Question 2.4:
Which variable instances have information flow over (last,16)? List all such variables.
  Output:
```json
{
"InfomationFlowSources": [
["arr", 10],
["arr", 8],
["arr", 3],
["j", 14],
["j", 5],
["size", 2],
["i", 4],
["i", 15],
["val", 1]
]
}
```
[YOUR TURN]
Below is **your target snippet**.
```
<target code with line number>
```
**Question**: Which variable instances have information flow over (dst_var, dst_line) in function target_function_name?
List all such variables.
**Output**:

