# OpenReview forum: "CoRe: Benchmarking LLMs’ Code Reasoning Capabilities through Static Analysis Tasks"
_NeurIPS.cc/2025/Datasets_and_Benchmarks_Track — NeurIPS 2025 Datasets and Benchmarks Track spotlight_

### Official Review · Reviewer_4X3h · 2025-06-16

**Rating:** 5
**Confidence:** 3

**Summary:**

This paper introduces CoRe, a benchmark specifically designed to evaluate large language models (LLMs) on fundamental static code analysis tasks—data dependency, control dependency, and information flow—across C/C++, Java, and Python. The benchmark consists of 12,564 human-verified task instances from real-world programs.

**Additional Feedback:**

None.

**Dataset Code Accessibility:**

Yes

**Dataset Code Comments:**

The dataset is available at https://huggingface.co/datasets/researchartifact1234/CoRe and https://github.com/researchartifact1234/CoRe

**Ethical Considerations:**

No, there are no or only very minor ethics concerns

**Final Justification:**

Good paper. I will maintain my score.

**Limitations Weaknesses:**

As the author points out, the benchmark is restricted to functions under 100 lines of code, which may not fully represent real-world software systems where inter-procedural reasoning and larger contexts are critical.
The focus on intra-procedural analysis overlooks tasks that involve multiple functions or modules, which are essential for many static analysis applications.

**Strengths Contributions:**

CoRe addresses a critical evaluation gap in code reasoning by focusing on semantic properties, rather than end-to-end task performance.
The benchmark contains rigorously annotated and human-verified instances from real-world code bases, ensuring authenticity.
Ten state-of-the-art LLMs are benchmarked using precise metrics (F1, Correct Trace Rate, and Exact Match).
This paper systematically identifies performance bottlenecks, such as control complexity, function length, and backward dependencies

---

> ### Author Rebuttal · Authors · 2025-07-29
>
> We are grateful for the thorough review and constructive feedback. Our responses to the comment is detailed below:
>
> ---
> > As the author points out, the benchmark is restricted to functions under 100 lines of code, which may not fully represent real-world software systems where inter-procedural reasoning and larger contexts are critical. The focus on intra-procedural analysis overlooks tasks that involve multiple functions or modules, which are essential for many static analysis applications.
>
>
> Thanks for pointing this out. We agree. However, annotating functions longer than 100 lines is difficult due to the challenges of automation (Appendix B1). To ensure high-quality annotations, we limit the benchmark to functions under 100 LoC. We respectfully argue that functions within this size range still **reflect real-world practices**, as well-structured codebases often keep functions concise for maintainability (e.g., ~40–50 LoC) [1-2].
>
> We will consider supporting inter-procedural analysis as future work. Despite the difficulty of obtaining reliable annotations, one possible approach is to use lightweight static analysis tools to identify function pairs with dependencies (e.g., via calling context). If each function is fully annotated, we can then construct a call graph to support inter-procedural reasoning.
>
> ----
> **References:**
>
> [1] https://google.github.io/styleguide/cppguide.html#Write_Short_Functions
>
> [2] https://docs.nvidia.com/legate/25.07/developer/style.html#functions

---

> > ### Comment · Reviewer_4X3h · 2025-08-04
> >
> > Thank you for your responses. I would keep my rating.

---

> ### Author Response · Authors · 2025-08-04
>
> Thanks for your time and consideration. We appreciate your support!

---

### Official Review · Reviewer_C139 · 2025-06-17

**Rating:** 5
**Confidence:** 3

**Summary:**

This paper proposes a new benchmark dataset aimed at evaluating the code reasoning capabilities of large language models (LLMs). Unlike previous benchmarks, this work emphasizes the models' ability to understand code dependencies and information flow. The authors aim to isolate and measure reasoning skills essential for more complex programming tasks.

**Dataset Code Accessibility:**

Yes

**Ethical Considerations:**

No, there are no or only very minor ethics concerns

**Final Justification:**

Good paper, accept.

**Limitations Weaknesses:**

1. The connection between reasoning ability and real-world coding performance remains unclear. It would strengthen the paper to demonstrate whether poor performance on the benchmark correlates with failures on realistic downstream tasks.

2. The metrics used in Figure 3 are inconsistent: the first two figures report correct trace rates, while the third and fourth use exact match and F1 score, respectively. The authors should justify or standardize the evaluation criteria.

3. The comparison with existing benchmarks is limited. In particular, the paper should more thoroughly compare with:

    -  **CRUXEval: A Benchmark for Code Reasoning, Understanding, and Execution**

    - **CodeMMLU: A Multi-Task Benchmark for Assessing Code Understanding & Reasoning Capabilities of CodeLLMs**

**Strengths Contributions:**

1. The paper is well-written, with clear logic and fluent language throughout.

2. The motivation is well-articulated, and the research problem is timely and relevant.

3. The proposed reasoning tasks are meaningful and help highlight a core aspect of code understanding.

4. The authors conduct a comprehensive evaluation of 10 LLMs on the benchmark, providing useful insights into current model capabilities.

---

> ### Author Rebuttal · Authors · 2025-07-29
>
> Thank you for your valuable feedback! We included the response to each point below:
>
> ---
>
> > The connection between reasoning ability and real-world coding performance remains unclear. It would strengthen the paper to demonstrate whether poor performance on the benchmark correlates with failures on realistic downstream tasks.
>
> As discussed in Sections 1–2, fundamental static analysis skills are essential for many downstream tasks such as program repair and test generation, which are core use cases for LLMs today. More importantly, recent work [1][2] has begun to directly use LLMs as static analyzers. These reasoning abilities also correlate with real-world coding performance. For example, public leaderboards on coding tasks [3-6] show similar trends to our results, with models like GPT‑o3 and Gemini 2.5 Pro consistently leading across both reasoning and practical coding tasks, with DeepSeek V3/R1 and Claude 3.7 also among the top performers.
>
> ---
>
> > The metrics used in Figure 3 are inconsistent: the first two figures report correct trace rates, while the third and fourth use exact match and F1 score, respectively. The authors should justify or standardize the evaluation criteria.
>
> Thanks for pointing this out! We use different metrics to cover all three tasks: F1 score for dependency classification, correct trace rate for trace prediction, and exact match rate for source enumeration. Due to space constraints, we show 1-2 sub-figures per task in Figure 3, but the observed trends hold across others as well. We will consider adding additional figures in the appendix for completeness.
>
> ---
>
> > The comparison with existing benchmarks is limited. In particular, the paper should more thoroughly compare with: “CRUXEval: A Benchmark for Code Reasoning, Understanding, and Execution” and “CodeMMLU: A Multi-Task Benchmark for Assessing Code Understanding & Reasoning Capabilities of CodeLLMs”
>
>
> CRUXEval[7] focuses **solely on Python** and evaluates input-output behavior, which aligns more with **dynamic analysis**. In contrast, our benchmark targets **static analysis** skills and covers Python, Java, and C/C++.
>
> For CodeMMLU[8], it includes two types of tasks: (1) general knowledge-based multiple-choice questions (e.g., how to run a Django server), and (2) fundamental tasks like code completion, repair, and execution prediction. As discussed in lines 41–51, like SweLancer and SweBench, these tasks evaluate models in an end-to-end manner, without isolating their reasoning over program semantics, which is an essential skill that CORE is designed to assess directly.
>
> Thanks for bringing these up! We will add them to our related work discussion.
>
> ----
> **References:**
>
> [1] Li, Ziyang, Saikat Dutta, and Mayur Naik. "IRIS: LLM-assisted static analysis for detecting security vulnerabilities." arXiv preprint arXiv:2405.17238 (2024).
>
> [2] Wang, Chengpeng, et al. "LLMDFA: analyzing dataflow in code with large language models." Advances in Neural Information Processing Systems 37 (2024): 131545-131574.
>
> [3] https://apxml.com/leaderboards/coding-llms
>
> [4] https://evalplus.github.io/leaderboard.html#:~:text=12,TestEval
>
> [5] https://lt-asset.github.io/REPOCOD/
>
> [6] https://www.swebench.com/index.html
>
> [7] Gu, Alex, et al. "Cruxeval: A benchmark for code reasoning, understanding and execution." arXiv preprint arXiv:2401.03065 (2024).
>
> [8] Manh, Dung Nguyen, et al. "CodeMMLU: A Multi-Task Benchmark for Assessing Code Understanding & Reasoning Capabilities of CodeLLMs." arXiv preprint arXiv:2410.01999 (2024).

---

> > ### Comment · Reviewer_C139 · 2025-08-07
> > **Thanks**
> >
> > Thank you for your responses. I would keep my rating.

---

> > > ### Author Response · Authors · 2025-08-07
> > >
> > > We appreciate your consideration and support!

---

### Official Review · Reviewer_cJNu · 2025-07-14

**Rating:** 5
**Confidence:** 5

**Summary:**

This paper introduces a multilingual benchmark designed to evaluate LLMs on fundamental static code analysis tasks. The benchmark contains large number of human-verified instances across C/C++, Java, and Python. It emphasizes semantic reasoning rather than surface-level pattern recognition, distinguishing itself from end-to-end evaluation benchmarks. The results reveal that while models do well on simple dependency classification, they perform poorly on trace generation and dependency enumeration, especially in complex control structures or reverse dependency settings.

**Additional Feedback:**

(1) How well do models handle pointer aliasing and array index tracking in C/C++ examples? These often lead to transitive and non-trivial data dependencies.

(2) Did the authors consider ablation studies for the sampling method (e.g., what if you drop semantic-aware heuristics)?

(3) Can the dataset be extended to support retrieval-augmented evaluation, where relevant examples are provided? The paper hints at this but leaves open how to best retrieve semantically aligned exemplars. How about tool usage (like how a typical software programmer does such analysis)?

(4) Are fine-tuned models on static analysis tasks able to outperform generic reasoning LLMs? Would be valuable to see baselines from models trained specifically on code semantics.

(5) How sensitive is the benchmark to syntactic variations in the code (e.g., renaming variables, code formatting, unrolling loops)?

**Dataset Code Accessibility:**

Yes

**Ethical Considerations:**

No, there are no or only very minor ethics concerns

**Limitations Weaknesses:**

-- While the benchmark is rigorous and well-constructed, it does exhibit some limitations. First, the current version is restricted to intra-procedural analysis, excluding inter-procedural dependencies such as those involving function calls or shared global state. This constraint is acknowledged by the authors but limits the benchmark’s applicability to real-world software, where cross-function reasoning is often critical. Although not necessarily a flaw, especially given the goal of isolating core semantic reasoning, it would be valuable for future versions to expand toward inter-procedural tasks, especially those where LLMs perform poorly (<10% accuracy), which could help guide model improvements.

--  Although the paper does a good job of quantifying performance, there is minimal discussion of annotation error rates or ambiguity in dependency labeling. The authors mention an 87.5% agreement rate among annotators, but it remains unclear how disagreements were resolved and how often complex or ambiguous cases arose. For example, implicit flows due to control conditions can sometimes be debatable depending on interpretation granularity.

-- The evaluation focuses only on deterministic inference settings (temperature = 0). While this avoids randomness, it also sidesteps issues of consistency and variability in model predictions, especially relevant in open-ended reasoning tasks like trace generation or enumeration. I am not sure if you have included such studies in the Appendix.

-- Another open question is how well the benchmark tasks generalize across programming paradigms. While the benchmark includes three languages. It remains to be seen whether these results translate to functional languages, DSLs, or newer hybrid paradigms. Especially, for the languages for which their data is not representative in the training set.

**Strengths Contributions:**

++ Improving benchmarking of LLMs for code reasoning by shifting the evaluation focus from end-task correctness to semantic understanding of program behavior.

++ Real-world tasks like vulnerability detection, debugging, and program repair require fine-grained reasoning over code semantics, which CORE directly targets, provide a well-motivated problem. The distinction from prior work is clearly articulated, esp. compared to popular code-related benchmarks like SWE-Bench.

++ The benchmark covers three widely used programming languages (C/C++, Java, Python). Additionally, it includes pairwise dependency queries and enumeration tasks, capturing both localized and holistic reasoning challenges. This is further enhanced by the semantics-aware diverse sampling method that ensures structural diversity and reasoning complexity.

++ Tasks like source enumeration and trace prediction go beyond simple dependency recognition, demanding deeper analysis that current LLMs largely fail at (e.g., <50% correct trace rate for most models).

---

> ### Author Rebuttal · Authors · 2025-07-31
>
> We appreciate the reviewers' careful consideration of our work and their helpful feedback. Please find our responses to each comment below:
>
> ---
>
> > Limitation 1: Restricted to intra-procedural analysis, limiting applicability to real-world scenarios that require cross-function reasoning.
>
>
> Thanks for pointing this out. We agree with the limitation. We will consider supporting inter-procedural analysis as future work. Despite the difficulty of obtaining reliable annotations, one possible approach is to use lightweight static analysis tools to identify function pairs with dependencies (e.g., via calling context). If each function is fully annotated, we can then construct a call graph to support inter-procedural reasoning.
>
> ---
> > Limitation 2: Lacks detailed discussion of annotation disagreements and ambiguity, despite reporting an 87.5% agreement rate.
>
> To resolve annotation disagreements, we strictly followed the definitions provided in Section 2. The 12.5% (100 - 87.5%) disagreement cases were typically due to ambiguity, which we addressed through discussion among all authors and iterative refinement to ensure consistency across all annotations. We will add more discussion to the paper.
>
> ---
>
> > Limitation 3: Evaluates only under deterministic inference (temperature = 0), leaving variability and consistency under different decoding settings unexplored.
>
> We acknowledge the reviewer's point regarding deterministic inference (temperature=0). For reasoning tasks, this is standard practice [1-4]. Evaluating with temperature > 0 introduces variability, making single-run results unreliable and evaluation challenging. Ensuring self-consistency would necessitate multiple runs, significantly increasing the token cost. More importantly, our primary contribution, however, lies in providing a robust, high-quality dataset that enables consistent and reproducible benchmarking, which is a distinct focus from exploring model variability under non-deterministic settings.
>
> ---
>
> > Limitation 4: Generalizability to other programming paradigms (e.g., functional languages or DSLs) remains unaddressed.
>
> This is a valuable point. Our current benchmark focuses on mainstream imperative and object-oriented languages (C/C++, Java, Python). While we acknowledge the importance of evaluating functional languages, DSLs, and hybrid paradigms, this is beyond the scope of the current work. Future extensions of CoRe could include such programming languages as Ruby, Ada, or Haskell to broaden its generalizability.
>
> ---
>
> > (1) How well do models handle pointer aliasing and array index tracking in C/C++ examples? These often lead to transitive and non-trivial data dependencies.
>
> We acknowledge the reviewer's insightful question regarding the models' ability to handle pointer aliasing and array index tracking in C/C++ examples. These constructs indeed introduce significant complexity in data dependency analysis.
>
> To investigate this, we compared the performance of Claude 3.5 on C/C++ tasks derived from programs containing pointer/array usage against the overall results. The data is presented below:
>
> **Claude 3.5 Performance Comparison (Overall vs. Programs with Pointer/Array Usage)**
>
> | Metric | Overall | Programs with Pointer/Array Usage |
> |---|---|---|
> | InfoFlow F1 for Dependency Classification | 82.84 | 82.82 |
> | InfoFlow Correct Trace Rate for Trace Generation | 31.27 | 30.88 |
> | DataDep F1 for Dependency Classification | 67.64 | 67.06 |
> | DataDep Correct Trace Rate for Trace Generation | 50.39 | 49.80 |
>
> As observed, there is a consistent, albeit small, drop in performance across all metrics when evaluating programs involving pointer/array usage. This suggests that models do find these cases more challenging. However, the relatively small performance gap also indicates that the models exhibit a degree of robustness, or that the presence of such constructs in a file does not always directly translate to the specific task instance requiring deep reasoning about them.
>
> ---
> > (2) Did the authors consider ablation studies for the sampling method (e.g., what if you drop semantic-aware heuristics)?
>
> We appreciate this insightful question regarding ablation studies for our sampling method. As detailed in Section 3.2.2, our semantics-aware sampling strategy is crucial for ensuring reasoning complexity. To investigate the impact of this, we conducted an experiment where we removed all heuristics and randomly sampled 400 task instances for the data dependency task.
>
> The performance of Claude 3.5 on this randomly sampled dataset, compared to our original settings, is shown below:
>
> **F1 for Dependency Classification**
>
> | Language | Original | Random Sampling |
> |---|---|---|
> | C | 82.84 | 83.86 |
> | Java | 82.63 | 85.78 |
> | Python | 81.61 | 85.06 |
>
> **Correct Trace Rate for Trace Prediction**
>
> | Language | Original | Random Sampling |
> |---|---|---|
> | C | 31.27 | 37.14 |
> | Java | 31.27 | 37.67 |
> | Python | 29.29 | 31.25 |
>
> As observed, performance is consistently better with random sampling. This indicates that random sampling tends to yield easier tasks, as it does not enforce the same level of reasoning complexity as our semantics-aware approach.
>
> It is important to note that different sampling strategies result in completely different datasets, and thus, a direct comparison of model performance across these datasets does not necessarily reflect a model's inherent ability.
>
> ----
> > (3) Can the dataset be extended to support retrieval-augmented evaluation, where relevant examples are provided? The paper hints at this but leaves open how to best retrieve semantically aligned exemplars. How about tool usage (like how a typical software programmer does such analysis)?
>
> Our dataset, comprising code and corresponding annotations, is designed to be highly flexible and can indeed support such advanced evaluation paradigms. As discussed in Section 5.3 and Appendix H, we have already conducted few-shot learning (FSL) experiments, demonstrating one way to leverage the dataset beyond zero-shot evaluation.
>
> Our provided prompt (Appendix I) serves merely as an illustrative example of how to interact with models using our dataset. Users are free to customize their prompting strategies, including incorporating RAG by inserting relevant examples or contexts, or even integrating tool usage. The dataset's utility is not limited to a specific evaluation methodology; rather, it provides the foundational data for diverse and innovative benchmarking approaches.
>
> Thanks for pointing this out. We will clarify this in the paper.
>
> ---
> > (4) Are fine-tuned models on static analysis tasks able to outperform generic reasoning LLMs? Would be valuable to see baselines from models trained specifically on code semantics.
>
>
> While fine-tuning smaller models (e.g., 1B parameters) on complex coding tasks requiring reasoning has shown limited performance, often not comparable to state-of-the-art reasoning or MoE models. This is evidenced by the substantial performance gap on SWE-bench[5], where a smaller model, Swe-llama 13B fine-tuned on the task, achieves 1% resolution rates, while SOTA systems reach over 60%. On the other hand, fine-tuning larger models (e.g., 70B parameters) is resource-intensive and beyond the scope of this work.
>
> Our primary contribution is that we provide a high-quality dataset. This dataset serves as a valuable resource for future research, enabling the community to explore various fine-tuning strategies and evaluate their effectiveness on code semantics tasks. Our benchmark is designed to facilitate such investigations, rather than being limited to specific fine-tuning approaches.
>
> ---
> > (5) How sensitive is the benchmark to syntactic variations in the code (e.g., renaming variables, code formatting, unrolling loops)?
>
> We appreciate the thoughtful question. While this is **orthogonal** to our current focus, our dataset is compatible with such studies. In a small-scale test, we manually refactored 10 samples (e.g., renaming `N` to `num`, `len` to `size`) and observed minimal performance differences, with fluctuations within 0.02%. There is existing systematic research on this topic[5][6], and we agree it’s an important direction for future exploration.
>
> ---
> **References:**
>
> [1] Ye, Xi, et al. "Satlm: Satisfiability-aided language models using declarative prompting." Advances in Neural Information Processing Systems 36 (2023): 45548-45580.
>
> [2] Sprague, Zayne, et al. "To cot or not to cot? chain-of-thought helps mainly on math and symbolic reasoning." arXiv preprint arXiv:2409.12183 (2024).
>
> [3] Wang, Chengpeng, et al. "LLMDFA: analyzing dataflow in code with large language models." Advances in Neural Information Processing Systems 37 (2024): 131545-131574.
>
> [4] Wang, Chengpeng, et al. "Sanitizing large language models in bug detection with data-flow." Findings of the Association for Computational Linguistics: EMNLP 2024. 2024.
>
> [5] https://www.swebench.com/index.html
>
> [6] Zhang, Yuhao, et al. "CodeFort: Robust training for code generation models." arXiv preprint arXiv:2405.01567 (2024).
>
> [7] Wang, Shiqi, et al. "ReCode: Robustness evaluation of code generation models." arXiv preprint arXiv:2212.10264 (2022).

---

### Official Review · Reviewer_xmmn · 2025-07-18

**Rating:** 5
**Confidence:** 4

**Summary:**

This submission introduces CoRe, a benchmark designed to evaluate LLMs on static analysis tasks, specifically on dependency reasoning. The benchmark is multilingual. Task instances are derived from manually annotated and human-verified programs using a semantics-aware sampling strategy to ensure both diversity and complexity. The authors evaluate 10 SOTA LLMs, including both reasoning and non-reasoning models. The result shows that while models generally perform well on classification task, trace generation/source enumeration remain a challenge.

**Additional Feedback:**

1. what types of error do LLMs make for trace generation/source enumeration. e.g. Are they missing steps or generating wrong edges? While some of this analysis are available in the appendix, it's worth discussing in the main text. Further, a comparison between LLMs could be interesting, e.g. do they make similar mistakes?
2. The paper reports that LLMs struggle with reverse dependency reasoning/early exit. Have you tried incorporating such cases explicitly into the base prompt? The base prompt included in the appendix seems very long. Have you tried shorten the prompt? Moreover, beyond the three factors discussed (function length, control structures, reverse dependencies), are there other interesting patterns or failure modes worth highlighting?
3. The conclusion could be strengthened by briefly discussing future directions.

**Dataset Code Accessibility:**

Yes

**Dataset Code Comments:**

The dataset and code are available.

**Ethical Comments:**

The authors explained the data collection process and identified potential bias in this work. There's no significant ethical concerns remain.

**Ethical Considerations:**

No, there are no or only very minor ethics concerns

**Final Justification:**

The work is solid and the authors addressed my concerns in the rebuttal. Final rating: accept.

**Limitations Weaknesses:**

It would be helpful to provide more insights/error analysis on the types of errors LLMs make in these tasks.

**Strengths Contributions:**

1. The paper is well-written and clearly motivated.

2. The task formulations and evaluation metrics are appropriate and well-justified.

3. The experiment is comprehensive.

---

> ### Author Rebuttal · Authors · 2025-07-29
>
> We are sincerely grateful for the insightful comments and valuable suggestions. Our point-by-point responses follow:
>
> ----
>
> > what types of error do LLMs make for trace generation/source enumeration. e.g. Are they missing steps or generating wrong edges?  While some of this analysis are available in the appendix, it's worth discussing in the main text. Further, a comparison between LLMs could be interesting, e.g. do they make similar mistakes?
>
>
> We appreciate the question regarding the types of errors LLMs make in trace generation and source enumeration. Our detailed results in Tables 9 and 10 provide insights into these error patterns.
>
> For trace generation (Table 9), we observe that information flow tasks consistently exhibit significantly higher average missing steps (Miss.) and invalid edge (IE) rates compared to data and control dependency task types. This suggests that information flow is the most challenging task for LLMs, where they frequently generate incorrect edges. A comparison between Claude 3.5 and Claude 3.7 reveals that while their valid edge rates (VE) are similar, Claude 3.7 demonstrates a much lower invalid edge rate (IE). This indicates Claude 3.7 possesses better reasoning capabilities, leading to fewer erroneous edges.
>
>
> Regarding dependency source enumeration (Table 10), different models display distinct error tendencies. For models such as Claude 3.7, DeepSeek R1, Llama 3.1 (405B), Qwen 3 (235B), GPT 4-mini, and GPT 4o, recall is consistently lower than precision across all task types (data dependency, control dependency, and information flow). This pattern suggests these models tend to yield precise dependency sources but frequently miss many relevant ones. Conversely, models like Gemini 2.5 Pro and DeepSeek V3 do not exhibit such a consistent trend in their precision-recall balance.
>
> Thank you for this valuable suggestion; we will incorporate this detailed discussion into the paper.
>
> ------
>
> > The paper reports that LLMs struggle with reverse dependency reasoning/early exit. Have you tried incorporating such cases explicitly into the base prompt?
>
> We have explored this approach by adding an example of reverse dependency reasoning to the prompt and conducting experiments with Claude 3.5 on data dependency tasks.
>
> **F1 for Dependency Classification**
>
> | Language | Original | Modified Prompt |
> |---|---|---|
> | C | 67.64 | 66.86 |
> | Java | 68.14 | 68.34 |
> | Python | 75.04 | 74.60 |
>
> **Correct Trace Rate for Trace Generation**
>
> | Language | Original | Modified Prompt |
> |---|---|---|
> | C | 50.39 | 50.00 |
> | Java | 44.86 | 45.57 |
> | Python | 42.30 | 39.32 |
>
> The results show negligible differences with no significant improvement or decrease. However, our primary contribution is providing a robust dataset rather than optimizing model performance through prompt engineering. Including examples or things to pay attention to for all challenging scenarios would be impractical and could be considered a form of overfitting to specific cases rather than evaluating genuine reasoning capabilities.
>
> ------
>
>
> > The base prompt included in the appendix seems very long. Have you tried shorten the prompt?
>
> We understand the concern regarding the length of the base prompt. Our prompts are designed to be concise while including necessary definitions and illustrative examples, which are particularly important for non-reasoning models and align with common few-shot learning practices. Modern LLMs also support very long input contexts, making prompt length less of a constraint.
>
> To investigate the impact of prompt length, we conducted an experiment where we shortened the prompts by cutting half of the examples. The results for Claude 3.5 on data dependency tasks are as follows:
>
> **F1 for Dependency Classification**
>
> | Language | Original | Shortened |
> |---|---|---|
> | C | 67.64 | 67.50 |
> | Java | 68.14 | 68.17 |
> | Python | 75.04 | 74.04 |
>
> **Correct Trace Rate for Trace Generation**
>
> | Language | Original | Shortened |
> |---|---|---|
> | C | 50.39 | 49.80 |
> | Java | 44.86 | 46.63 |
> | Python | 42.30 | 39.86 |
>
> These results show negligible overall differences, with a slight decrease in some metrics.
>
> ------
>
> > Moreover, beyond the three factors discussed (function length, control structures, reverse dependencies), are there other interesting patterns or failure modes worth highlighting?
>
> Certainly, beyond the three factors discussed, we have identified other patterns that influence model performance. For example, maximum control flow depth and the path complexity within the control flow graph (CFG) are also significant. A greater control dependence path length between two variables often correlates with a higher error rate. In the paper, we focused on the more straightforward factors in our initial analysis for clarity, but these more nuanced structural properties are a promising direction for future investigation. We will include more analyses in the appendix.
>
>
> > The conclusion could be strengthened by briefly discussing future directions.
>
> Thank you for the valuable suggestion! For future work, we will consider supporting inter-procedural analysis. Despite the difficulty of obtaining reliable annotations, one possible approach is to use lightweight static analysis tools to identify function pairs with dependencies (e.g., via calling context). If each function is fully annotated, we can then construct a call graph to support inter-procedural reasoning.
>
> Another promising direction is to investigate the resilience of LLMs' code reasoning to various forms of code obfuscation or subtle adversarial perturbations. This would bring insights into the robustness and reliability of their semantic understanding in challenging real-world scenarios.

---

> > ### Comment · Reviewer_xmmn · 2025-08-05
> >
> > Thank you for addressing my concerns and conducting the additional experiments. I have no further questions and will keep the rating. Please incorporate the additional analysis to the paper. Look forward to the final version!

---

> > > ### Author Response · Authors · 2025-08-05
> > >
> > > Thanks for the time and consideration. We appreciate your support!

---

### Note · Authors · 2025-08-15

We sincerely thank the reviewers and ACs for their constructive feedback and thoughtful engagement. We are encouraged by the consensus that our work is technically solid, well-motivated, and impactful for advancing evaluation of LLMs on semantic code reasoning.

Reviewers consistently highlighted strengths: (i) the benchmark’s clear motivation and novelty in isolating semantic reasoning skills, (ii) rigorous task design and semantics-aware sampling, (iii) comprehensive experiments across 10 state-of-the-art LLMs, and (iv) high-quality human-verified annotations.

We also carefully addressed key concerns:

- **Additional analysis**: We added detailed discussions and analyses of error types, contrasted failure patterns across models, addressed the challenges of pointer aliasing and array index tracking (particularly in C/C++), conducted an ablation study on the semantic-aware sampling strategy, examined sensitivity to syntactic variations, and explored complexity measures beyond function length, control structures, and reverse dependencies.
- **Prompting strategies**: We experimented with shorter prompts and explicit reverse-dependency cases. Results showed negligible performance changes, confirming that our benchmark’s primary contribution is providing a robust evaluation foundation rather than prompt tuning.
- **Limitations and scope**: We focus on intra-procedural reasoning, with a ≤100 LoC cap chosen for annotation quality and reflecting common real-world practice. Inter-procedural reasoning is a natural next step for future work.
- **Connections to real-world tasks and related work**: We clarified how our benchmark underpins downstream applications (e.g., program repair, vulnerability detection) and distinguished it from CRUXEval, CodeMMLU. We also discussed its applicability in evaluating RAG systems and FSL scenarios.
- **Clarifications:** We detailed the annotation disagreement resolution process for the 12.5% conflicting cases, justified the task-specific metric choices in Figure 3, and stated explicitly that deterministic inference (T=0) was used to ensure reproducibility.

Importantly, we will integrate key reviewer suggestions into the final version. Our goal is to deliver a high-quality, semantics-aware benchmark that isolates reasoning skills and supports systematic exploration of prompting, fine-tuning, retrieval-augmented reasoning, and robustness.

We thank all reviewers for their positive feedback and constructive suggestions.

---

### Decision · Program_Chairs · 2025-09-18

**Decision:**

Accept (spotlight)

**Comment:**

This paper introduces CoRe, a new benchmark for evaluating the code reasoning capabilities of Large Language Models (LLMs) on fundamental static analysis tasks, specifically focusing on static analysis concepts like data dependency, control dependency, and information flow. The authors have done comprehensive evaluation on different LLMs for different coding languages. This work addresses a gap in the current evaluation landscape for code-focused LLMs.

Reviewers generally found the paper to be well motivated and of good coverage and comprehensiveness on the evaluation. The additional analysis provided by the authors during the rebuttal are also very helpful, and I see a solid revision after integrating these into the current draft.